# Geometric Mixture Models for Electrolyte Conductivity Prediction

**Anyi Li**[1 2 3]**, Jiacheng Cen**[1 2 3]**, Songyou Li**[1 2 3]**, Mingze Li**[1 2 3]**, Yang Yu**[4]**, Wenbing Huang**[1 2 3*]

[1] Gaoling School of Artificial Intelligence, Renmin University of China
[2] Beijing Key Laboratory of Research on Large Models and Intelligent Governance
[3] Engineering Research Center of Next-Generation Intelligent Search and Recommendation, MOE
[4] Hisun Pharm
{li_anyi, jiacc.cn, yangyubard}@outlook.com;
{songyou_li, hwenbing}@126.com; limingzetony@gmail.com;

## Abstract

Accurate prediction of ionic conductivity in electrolyte systems is crucial for advancing numerous scientific and technological applications. While significant progress has been made, current research faces two fundamental challenges: (1) the lack of high-quality standardized benchmarks, and (2) inadequate modeling of geometric structure and intermolecular interactions in mixture systems. To address these limitations, we first reorganize and enhance the CALiSol and DiffMix electrolyte datasets by incorporating geometric graph representations of molecules. We then propose GeoMix, a novel geometry-aware framework that preserves Set-SE(3) equivariance—an essential but challenging property for mixture systems. At the heart of GeoMix lies the Geometric Interaction Network (GIN), an equivariant module specifically designed for intermolecular geometric message passing. Comprehensive experiments demonstrate that GeoMix consistently outperforms diverse baselines (including MLPs, GNNs, and geometric GNNs) across both datasets, validating the importance of cross-molecular geometric interactions and equivariant message passing for accurate property prediction. This work not only establishes new benchmarks for electrolyte research but also provides a general geometric learning framework that advances modeling of mixture systems in energy materials, pharmaceutical development, and beyond.

## 1 Introduction

Predicting the properties of multi-molecular mixture systems plays a central role in advancing a wide range of emerging technologies, such as next-generation energy storage, advanced materials design, and pharmaceutical formulation [1–4]. Electrolytes, as a prototypical example of these systems, typically consist of lithium salts and organic solvents of diverse types. Accurate prediction of electrolyte conductivity is a key research focus, as it directly determines critical battery performance metrics including charge transfer efficiency, power density, and operational stability [5; 6]. However, modeling such complex systems presents significant challenges, as it requires simultaneously accounting for both the internal structure of individual molecules and the intricate interactions among different molecules. Classical theoretical approaches—ranging from the Nernst equation to dilute solution models—either neglect explicit intermolecular forces or rely on extrapolations from infinitely dilute regimes, resulting in limited accuracy and poor universality for practical applications [7; 8].

Spurred by recent advancements in machine learning, particularly deep learning, researchers have increasingly adopted learning-based approaches to model electrolyte systems [9–11], achieving

---

*Wenbing Huang is the corresponding author.

significant progress over classical methods. As a data-driven paradigm, the development of predictive models for these systems hinges critically on high-quality and standardized benchmarks. However, the field still faces substantial challenges: most existing datasets are semi-structured, requiring labor-intensive manual curation from primary literature due to inconsistent formatting standards. This lack of accessible, well-curated benchmarks has severely hindered systematic comparisons and reproducibility across methodologies.

Another critical limitation in existing research stems from inadequate geometric characterization of mixture systems. Previous works such as MM-MoLFormer [12] and MolSets [13] represent molecules in electrolyte systems using SMILES sequences or topological graphs. However, these approaches neglect the geometric structure of molecules in 3D space, which significantly impacts the properties of entire systems [14–17]. While recent methods such as DiffMix [11] and Uni-ELF [18] have incorporated 3D structural information, they aggregate molecular information into global features and simulate intermolecular interactions through comparing these aggregated representations, thereby overlooking fine-grained and full-atom interactions across molecules. A naive solution would involve direct atom-level coordinate transfer between molecules, but this violates the fundamental symmetries of mixture systems. Crucially, any physically meaningful representation must maintain consistency under: 1) independent permutation of each molecule's atoms; 2) independent SE(3) transformations (rotation/translation) of each molecule's local coordinate system; and 3) permutation of molecules within the system. We formalize these requirements as *Set-SE(3) equivariance* (see Eqs. (2) to (4)), a challenging but essential property for developing accurate models of molecular interactions.

To address the above two challenges, we first curate and standardize two public datasets for electrolyte conductivity prediction: CALiSol [19] and DiffMix [11]. Inspired by the powerful expressiveness of geometric information [20–44], we augment the molecular representation by constructing geometric graphs where atoms are characterized by both invariant features (*e.g.*, atomic numbers) and equivariant features (*e.g.*, 3D coordinates), ensuring downstream models can fully exploit 3D structural information. We then propose **GeoMix**, a geometry-aware framework for mixture system modeling and property prediction. At its core is the Geometric Interaction Network (GIN), a novel equivariant module designed for intermolecular geometric message passing. The fundamental idea involves first creating a local coordinate frame for each molecule, then computing learnable coordinate transformations to align atom pairs from different local frames, and finally performing equivariant message passing over the transformed coordinates. Our code and dataset are available at Github[2].

In summary, this paper contributes to the following aspects:

- We curate and standardize two public datasets, CALiSol and DiffMix, covering a wide range of electrolyte mixture systems. The datasets are carefully processed with geometric graph construction, forming a benchmark for electrolyte conductivity prediction.

- We propose GeoMix, a novel geometric framework for modeling mixture systems. GeoMix maintains Set-SE(3) equivariance while capturing fine-grained geometric relationships between molecules, overcoming the limitations of existing methods.

- We compare GeoMix against diverse baseline models built upon MLPs, GNNs, and geometric GNNs, under consistent settings. GeoMix consistently outperforms all baselines on both datasets, demonstrating its effectiveness in modeling mixture systems.

## 2 Related Work

**Early-Stage Methods** Earliest approaches compute ionic conductivity by applying the Nernst–Einstein equation, which assumes that all ions diffuse independently and intermolecular interactions can be neglected. Under this framework, each ionic species contributes additively to the total conductivity. While effective at low concentrations, this framework breaks down in concentrated or strongly interacting systems, where correlated ionic motion becomes significant [8]. With advances in computational chemistry, first-principles methods such as Density Functional Theory (DFT) [45] and Machine Learning Force Fields (MLFFs) [10; 46] became available. These techniques accurately capture interactions between ions and molecules over short timescales, enabling more precise estimates of ion mobilities under realistic intermolecular forces [10; 47–49]. Despite their accuracy, classical and ab initio MD simulations require modeling every atom in the local system, leading

---

[2]https://github.com/GLAD-RUC/GeoMix

to substantial computational expense. Moreover, because conductivity predictions rely on multiple intermediate steps, accumulated numerical errors can offset the benefits of high precision.

**Deep Learning Methods** Recent deep learning frameworks [11–13; 18; 50; 51] aim to model mixture systems directly from their components and proportions. These methods can be roughly grouped into three categories: (1) *SMILES-based methods* [12; 50; 51] compute mixture embeddings by weighted summation over molecular embeddings derived from molecular text representations such as SMILES. However, these approaches ignore the influence of molecular geometry on both the representation and interactions within the mixture. (2) *Topology-based methods* such as MolSets [13] model intermolecular interactions using self-attention over molecular-level graph embeddings. While molecular topology is captured, spatial interactions are only implicitly encoded through attention mechanisms. (3) *Geometry-based methods* further leverage 3D molecular information. DiffMix [11] aggregates molecular embeddings using learned polynomial mixing coefficients, while incorporating some physical priors. Uni-ELF [18] combines microscopic descriptors such as radial distribution functions (RDFs) with learned geometric features to extract invariant intermolecular representations. However, it does not capture equivariant features between molecules. While most studies about equivariant models focus on molecular interactions within a single reference frame [52; 53], our proposed method explicitly models node-level geometric relationships across molecules through equivariant message passing. This approach enables fine-grained, geometry-aware interaction modeling at the atomic scale.

# 3 Method

In this section, we first formulate the regression task for mixture systems and discuss its associated symmetry constraints in § 3.1. We then introduce GeoMix, a novel geometric learning framework designed for both scalar (invariant) and vector (equivariant) property prediction, in § 3.2. Finally, we show a crucial component of GeoMix, GIN, in § 3.3. All proofs are provided in Appendix A.

## 3.1 Preliminaries

**Notations** We employ three-level representations to describe mixture systems in a hierarchical way. 1) **System-level**: A mixture system is a set of $M$ molecules, *i.e.*, $\mathbb{S} = \{\mathcal{G}_m\}_{m=1}^M$, where the geometric graph $\mathcal{G}_m$ represents the $m$-th molecule in the system. The global environment vector $\boldsymbol{c}$ encodes physical conditions (*e.g.* temperature and pressure) and $\kappa$ denotes the target electrolyte conductivity. 2) **Graph-level**: Each geometric graph $\mathcal{G}_m(\boldsymbol{H}_m, \vec{\boldsymbol{X}}_m, \vec{\boldsymbol{V}}_m, w_m)$ contains $N_m$ nodes. The quantities $\boldsymbol{H}_m \in \mathbb{R}^{H \times N_m}, \vec{\boldsymbol{X}}_m \in \mathbb{R}^{3 \times N_m}$ denote node scalar features and node 3D coordinates, respectively; $\vec{\boldsymbol{V}}_m \in \mathbb{R}^{3 \times N_m}$ refers to node geometric features, which are usually initialized as zeros or just node coordinates $\vec{\boldsymbol{X}}_m$; $0 < w_m < 1$ is the proportion of molecule $\mathcal{G}_m$ in the mixture, and the sum of all $w_m$s is equal to 1. All coordinates $\vec{\boldsymbol{X}}_m$ are centered at the origin beforehand, implying that the model should be translation-invariant. 3) **Node-level**: We denote by $\boldsymbol{h}_i^{(m)} \in \mathbb{R}^H, \vec{\boldsymbol{x}}_i^{(m)}, \vec{\boldsymbol{v}}_i^{(m)} \in \mathbb{R}^3$ the representations of the $i$-th node in $\mathcal{G}_m$.

**Task Definition** We aim to predict both scalar and geometric properties of the mixture system $\mathbb{S}$. Formally, our goal is to learn a mapping $\varphi$ from system space to property space:

$$\varphi \colon (\{\mathcal{G}_m(\boldsymbol{H}_m, \vec{\boldsymbol{X}}_m, \vec{\boldsymbol{V}}_m, w_m)\}_{m=1}^M, \boldsymbol{c}) \mapsto (\{\boldsymbol{H}_m', \vec{\boldsymbol{V}}_m'\}_{m=1}^M, \kappa), \tag{1}$$

where $\boldsymbol{H}_m'$ and $\vec{\boldsymbol{V}}_m'$ respectively represent the predicted scalar and geometric properties of all nodes. These node-level properties can be easily aggregated via readout functions to derive graph-level or system-level property $\kappa$, such as the electrolyte conductivity.

**Symmetries** Symmetries are fundamental in the physical world and should be satisfied when modeling interactions between physical systems. In mixture systems, symmetries are more complicated than those in single-instance systems. To be specific, the symmetries of the function $\varphi$ in Eq. (1) are divided into two parts: the *node-level* and the *graph-level* symmetry constraints.

For node-level symmetries, the output should be permutation-equivariant within each graph, namely,

$$\varphi \colon (\{\mathcal{G}_m(\boldsymbol{H}_m \boldsymbol{P}_m, \vec{\boldsymbol{X}}_m \boldsymbol{P}_m, \vec{\boldsymbol{V}}_m \boldsymbol{P}_m, w_m)\}_{m=1}^M, \boldsymbol{c}) \mapsto (\{\boldsymbol{H}_m' \boldsymbol{P}_m, \vec{\boldsymbol{V}}_m' \boldsymbol{P}_m\}_{m=1}^M, \kappa), \tag{2}$$

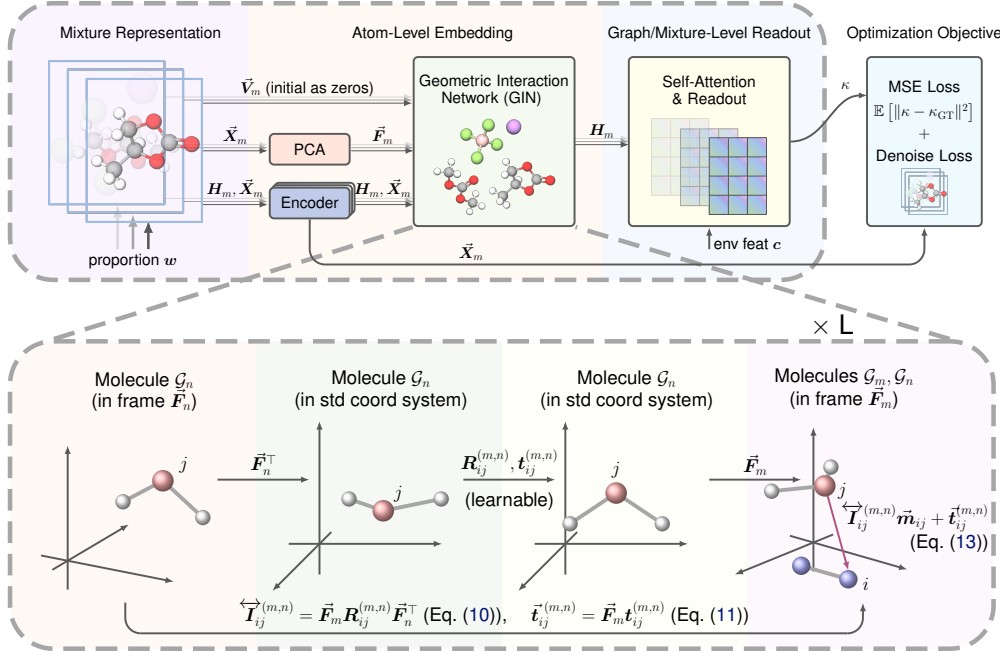

Figure 1: Overview of our GeoMix. By taking a set of molecules $\{\mathcal{G}_m\}_{m=1}^M$ with proportion $\boldsymbol{w} = [w_m]_M$ as input, it first constructs local frames $\vec{\boldsymbol{F}}_m$ via PCA, then applies an equivariant encoder to update $\boldsymbol{H}_m, \vec{\boldsymbol{X}}_m$. Intermolecular message passing is performed via GIN, which learns transformations $\overleftrightarrow{\boldsymbol{I}}_{ij}^{(m,n)}$ and $\vec{\boldsymbol{t}}_{ij}^{(m,n)}$, which enable equivariant message passing across molecules. The scalar features $\boldsymbol{H}_m$ are finally aggregated along with the environment descriptor $\boldsymbol{c}$ for prediction.

where $\boldsymbol{P}_m \in \mathbb{R}^{N_m \times N_m}$ is a permutation matrix acting all nodes in graph $\mathcal{G}_m$. Besides, the output should also be rotation-equivariant on each graph, *i.e.*,

$$\varphi \colon \left(\{\mathcal{G}_m(\boldsymbol{H}_m, \boldsymbol{R}_m \vec{\boldsymbol{X}}_m, \boldsymbol{R}_m \vec{\boldsymbol{V}}_m, w_m)\}_{m=1}^M, \boldsymbol{c}\right) \mapsto \left(\{\boldsymbol{H}_m', \boldsymbol{R}_m \vec{\boldsymbol{V}}_m'\}_{m=1}^M, \kappa\right), \tag{3}$$

where $\boldsymbol{R}_m \in \mathbb{R}^{3 \times 3}$ is a rotation matrix acting on node coordinates and geometric features in $\mathcal{G}_m$. Translation equivariance is naturally achieved by centering each graph, and is therefore omitted.

For graph-level symmetries, the output should be equivariant with respect to the order of the graphs:

$$\varphi \colon \left(\{\mathcal{G}_{\sigma(m)}\}_{m=1}^M, \boldsymbol{c}\right) \mapsto \left(\{\boldsymbol{H}_{\sigma(m)}', \vec{\boldsymbol{V}}_{\sigma(m)}'\}_{m=1}^M, \kappa\right), \tag{4}$$

where $\sigma$ denotes any permutation acting on all graphs $\{\mathcal{G}_m\}_{m=1}^M$.

Note that the output property $\kappa$ is invariant to any transformation. We call the two-level symmetries as **Set-SE(3) Equivariance** in this paper.

### 3.2 Model Architecture

As illustrated in Fig. 1, we design a general equivariant architecture that enables the update of both invariant and equivariant features through intramolecular and intermolecular message passing.

**Proportion Embedding** Each molecule in the mixture system is associated with a concentration value $w_m$. Prior studies [13; 50] often treat each molecule as having a fixed representation regardless of its concentration. In chemical thermodynamics, however, the activity of each molecule depends nonlinearly on its mole fraction and the activity coefficient, reflecting coupling between concentration and intermolecular forces [8]. To capture such context dependence, we concatenate each normalized proportion $w_i$ directly into atomic-level features via feature concatenation.

**Intramolecular Encoding** We employ either EGNN [54] or TFN [55] as equivariant encoders to characterize molecular geometries through message passing within individual molecules. The same encoder with shared parameters is used for all solvent molecules, while a separate encoder is

employed for salts due to their distinct structural properties (organic salts versus inorganic solvents). In this stage, we directly update node coordinates $\vec{\boldsymbol{X}}_m$ by treating them as geometric features $\vec{\boldsymbol{V}}_m$.

**Frame Construction**  The rotation equivariance described in Eq. (3) arises primarily because each molecule is represented in an independent local coordinate system. This inherent symmetry prevents direct transfer of coordinate information between different molecules. However, if we are able to derive the local frame of each molecule, we can conduct geometric message passing between different molecules by first transforming coordinate information into standard coordinate space, as depicted in Fig. 1. Here, we achieve this purpose based on Principal Component Analysis (PCA)[3]. In particular, we first compute the covariance $\mathrm{Cov}(\vec{\boldsymbol{X}}_m) = \frac{1}{N_m}\vec{\boldsymbol{X}}_m\vec{\boldsymbol{X}}_m^\top$, and then perform eigendecomposition $\mathrm{Cov}(\vec{\boldsymbol{X}}_m) = \vec{\boldsymbol{F}}_m\Lambda_m\vec{\boldsymbol{F}}_m^\top$ to obtain its orthogonal eigenvectors $\vec{\boldsymbol{F}}_m$ to form a local frame. If $\det(\vec{\boldsymbol{F}}_m) = -1$, we flip the third axis to ensure a right-handed coordinate system.

**Intermolecular Interaction**  With the pre-constructed local frames $\{\vec{\boldsymbol{F}}_m\}_{m=1}^M$, we design a novel component GIN to simulate geometric interactions between different molecules. The details of GIN will be presented in § 3.3. In this module, the coordinates $\vec{\boldsymbol{X}}_m$ are fixed, the geometric features $\vec{\boldsymbol{V}}_m$ are initialized as zeros and will be updated throughout GIN.

**Readout and Prediction Head**  The outputs of GIN are node-level features. For mixture property prediction tasks such as electrolyte conductivity prediction, we first average the node-level features and read them out as graph-level features, which are then modeled by a self-attention encoder without position encoding. The system-level feature is obtained by averaging these graph-level features, and finally, we concatenate with the global environment feature and pass the result through an MLP with an activation function to generate the prediction. These operations can be formulated as:

$$\boldsymbol{h}_m = \tfrac{1}{N_m}\mathbf{1}^\top\boldsymbol{H}_m, \quad y = \mathtt{MLP}\left(\tfrac{1}{M}\mathbf{1}^\top\mathtt{SelfAtt}\left(\bigoplus_{m=1}^M \boldsymbol{h}_m\right) \oplus \boldsymbol{c}\right), \tag{5}$$

where $\oplus$ represents the tensor concatenate operation, $y \in \mathbb{R}$ is the predicted conductivity.

**Optimization Objective**  For invariant prediction tasks, we use Mean Squared Error (MSE) as the loss function. To take full advantage of the equivariant network, we also adopt the Noisy Nodes method [57]. Noisy Nodes method encourages the model to learn the configurations with lower energy, effectively regularizing the learning process. Specifically, our model additionally learns a denoising process of the encoder:

$$\{\mathcal{G}(\boldsymbol{H}_m, \vec{\boldsymbol{X}}_m^{\mathrm{noisy}})\}_{m=1}^M \mapsto \{\mathcal{G}(\boldsymbol{H}_m, \vec{\boldsymbol{X}}_m^{\mathrm{denoise}})\}_{m=1}^M, \tag{6}$$

where $\vec{\boldsymbol{X}}_m^{\mathrm{noisy}} = \vec{\boldsymbol{X}}_m + \mathcal{N}(\mathbf{0}, \sigma^2\boldsymbol{I})$, and $\sigma = 0.3$ is taken for general molecular-level tasks. The auxiliary loss is calculated as:

$$\mathcal{L}_{\mathrm{denoise}} = \sum_{m=1}^M \|\vec{\boldsymbol{X}}_m^{\mathrm{denoise}} - \vec{\boldsymbol{X}}_m\|_{\mathrm{F}}, \tag{7}$$

where $\|\cdot\|_{\mathrm{F}}$ is the Frobenius norm. The optimization objective is the sum of two losses given by:

$$\mathcal{L} = \mathcal{L}_{\mathrm{MSE}} + \gamma \cdot \mathcal{L}_{\mathrm{denoise}}. \tag{8}$$

The implementation details can be found in Appendix B.

## 3.3  Geometric Interaction Network

In this section, we introduce GIN, a novel equivariant message-passing module to facilitate the geometric message passing between molecules using coordinate transformations, while promisingly ensuring Set-SO(3) equivariance.

**Architecture Overview**  GIN takes as input a set of geometric graphs $\{\mathcal{G}_m(\boldsymbol{H}_m, \vec{\boldsymbol{X}}_m, \vec{\boldsymbol{V}}_m)\}_{m=1}^M$ along with the corresponding local frames $\{\vec{\boldsymbol{F}}_m\}_{m=1}^M$. In each layer, the node features $\{\boldsymbol{H}_m\}_{m=1}^M$ and the geometry features $\{\vec{\boldsymbol{V}}_m\}_{m=1}^M$ are updated, while the coordinates $\{\vec{\boldsymbol{X}}_m\}_{m=1}^M$ are unchanged. To model the interaction from the source molecule $\mathcal{G}_m$ and the target one $\mathcal{G}_n$, GIN consists of three key components. 1) **Intermolecular Transformation**: Establishing a transformation within a

---

[3]Since frame construction via traditional PCA is not strictly equivariant owing to the non-unique axis-orientation [56], we use an improved, equivariance-preserving PCA variant (see Appendix B.8).

standard coordinate system based on the features of both the source and target molecules. 2) **Message Construction**: Computing the message in the source molecule's coordinate frame using the learned transformation, then converting it to the target molecule's frame. 3) **Aggregation and Update**: Aggregating the messages to update the node-level embeddings in the target molecule. For brevity, in our following formulation, $i \in \mathcal{G}_m$ consistently denotes the atom receiving message, while $j \in \mathcal{G}_n$ represents the atom sending message.

**Intermolecular Transformation**   Since the two molecules exist in different reference frames, the message passing strategy used in traditional equivariant GNNs is no longer suitable. Consequently, we aim to design a transformation that facilitates equivariant message passing across distinct reference frames. For atoms $i \in \mathcal{G}_m$ and $j \in \mathcal{G}_n$, we first calculate a scalar $z_{ij}^{(m,n)}$ to combine the invariant features in both atoms as follows:

$$z_{ij}^{(m,n)} = \sigma_{\text{inv}}(\boldsymbol{h}_i^{(m)}, \boldsymbol{h}_j^{(n)}, \|\vec{\boldsymbol{x}}_i^{(m)}\|, \|\vec{\boldsymbol{x}}_j^{(n)}\|, \|\vec{\boldsymbol{v}}_i^{(m)}\|, \|\vec{\boldsymbol{v}}_j^{(n)}\|), \tag{9}$$

where $\sigma_{\text{inv}}$ is an MLP, and subsequent $\sigma$ functions with different subscripts represent distinct MLPs. Based on $z_{ij}^{(m,n)}$ and the frames $\vec{\boldsymbol{F}}_m, \vec{\boldsymbol{F}}_n$, we introduce a learnable matrix $\overleftrightarrow{\boldsymbol{I}}_{ij}^{(m,n)} \in \mathbb{R}^{3\times3}$ to "rotate" the equivariant message from $j \in \mathcal{G}_n$ into the reference frame of $i \in \mathcal{G}_m$:

$$\overleftrightarrow{\boldsymbol{I}}_{ij}^{(m,n)} := \vec{\boldsymbol{F}}_m \boldsymbol{R}_{ij}^{(m,n)} \vec{\boldsymbol{F}}_n^\top, \quad \boldsymbol{R}_{ij}^{(m,n)} = \sigma_{\text{rot}}(z_{ij}^{(m,n)}) \in \mathbb{R}^{3\times3}. \tag{10}$$

Similarly, we also design a learnable vector $\vec{\boldsymbol{t}}_{ij}^{(m,n)}$ to model the "translation":

$$\vec{\boldsymbol{t}}_{ij}^{(m,n)} := \vec{\boldsymbol{F}}_m \boldsymbol{t}_{ij}^{(m,n)}, \quad \boldsymbol{t}_{ij}^{(m,n)} = \sigma_t(z_{ij}^{(m,n)}) \in \mathbb{R}^3. \tag{11}$$

It is worth noting that the matrix corresponding to $\overleftrightarrow{\boldsymbol{I}}_{ij}^{(m,n)}$ does not represent a strict rotation, as the determinant of $\boldsymbol{R}_{ij}^{(m,n)}$ is not constrained to be 1. We also explore alternative constructions of $\boldsymbol{R}_{ij}^{(m,n)}$ with a unit determinant, by using quaternion [58] and 6D vector [59]. However, ablation studies (see § 4.4) reveal that these alternatives yield less favorable results. In fact, we can give the following theorem to explain the expressive power of Eq. (10).

**Theorem 3.1** (Expressivity of Intermolecular Transformation Matrix). *Given geometric graphs $\mathcal{G}_m$ and $\mathcal{G}_n$, with $\vec{\boldsymbol{F}}_m, \vec{\boldsymbol{F}}_n$ being respective frames, any matrix $\overleftrightarrow{\boldsymbol{I}}_{mn}$ satisfying SO(3)-equivariance $\overleftrightarrow{\boldsymbol{I}}_{mn} \xmapsto{\boldsymbol{R}_m, \boldsymbol{R}_n \in \text{SO}(3)} \boldsymbol{R}_m \overleftrightarrow{\boldsymbol{I}}_{mn} \boldsymbol{R}_n^\top$ must take the form like Eq. (10) as $\overleftrightarrow{\boldsymbol{I}}_{mn}(\vec{\boldsymbol{F}}_m, \vec{\boldsymbol{F}}_n) = \vec{\boldsymbol{F}}_m \boldsymbol{R} \vec{\boldsymbol{F}}_n^\top$.*

**Message Construction**   The invariant message $\boldsymbol{m}_{ij}^{(m,n)}$ with geometric interaction is similar to the construction of $z_{ij}^{(m,n)}$ in Eq. (9). The difference is that in order to align the invariant message and the equivariant message, we additionally introduce the modulus of the translation vector $\vec{\boldsymbol{t}}_{ij}^{(m,n)}$ in Eq. (11). The entire formula is given by:

$$\boldsymbol{m}_{ij}^{(m,n)} = \sigma_{\text{msg}}(\boldsymbol{h}_i^{(m)}, \boldsymbol{h}_j^{(n)}, \|\vec{\boldsymbol{x}}_i^{(m)}\|, \|\vec{\boldsymbol{x}}_j^{(n)}\|, \|\vec{\boldsymbol{v}}_i^{(m)}\|, \|\vec{\boldsymbol{v}}_j^{(n)}\|, \|\vec{\boldsymbol{t}}_{ij}^{(m,n)}\|). \tag{12}$$

We now compute cross-reference frame equivariant messages by two steps. First, the atom coordinate $\vec{\boldsymbol{x}}_j^{(n)}$ and the geometric feature $\vec{\boldsymbol{v}}_j^{(n)}$ are combined to construct the message $\vec{\boldsymbol{m}}_{ij}^{(m,n)}$ in the reference frame of $\mathcal{G}_n$. Second, this message is transformed into the reference frame of $\mathcal{G}_m$ by applying the learnable transformation, *i.e.* the "rotation matrix" $\overleftrightarrow{\boldsymbol{I}}_{ij}^{(m,n)}$ and the "translation vector" $\vec{\boldsymbol{t}}_{ij}^{(m,n)}$, resulting in the updated message $\boldsymbol{m}_{ij}^{(m,n)}$ in the $\mathcal{G}_m$ frame:

$$\vec{\boldsymbol{m}}_{ij}^{(m)} = \overleftrightarrow{\boldsymbol{I}}_{ij}^{(m,n)} \vec{\boldsymbol{m}}_{ij}^{(m,n)} + \vec{\boldsymbol{t}}_{ij}^{(m,n)}, \quad \vec{\boldsymbol{m}}_{ij}^{(m,n)} = \sigma_{\vec{\boldsymbol{x}}}(\boldsymbol{m}_{ij}^{(m,n)})\vec{\boldsymbol{x}}_j^{(n)} + \sigma_{\vec{\boldsymbol{v}}}(\boldsymbol{m}_{ij}^{(m,n)})\vec{\boldsymbol{v}}_j^{(n)}. \tag{13}$$

**Aggregation and Update**   After devising the message, we proceed to aggregate them and update the atom features. Each atom is required to receive message from all atoms in every other molecule. To this end, we employ the average operator to aggregate both invariant and equivariant message:

$$\boldsymbol{m}_i^{(m)} = \frac{1}{N - N_m} \sum_{n \neq m} \sum_{j=1}^{N_n} \boldsymbol{m}_{ij}^{(m,n)}, \quad \vec{\boldsymbol{m}}_i^{(m)} = \frac{1}{N - N_m} \sum_{n \neq m} \sum_{j=1}^{N_n} \vec{\boldsymbol{m}}_{ij}^{(m)}, \tag{14}$$

where $N = \sum_{m=1}^{M} N_m$ denotes the number of all atoms in the system. Finally, we update the invariant feature $\boldsymbol{h}_i^{(m)}$ and equivariant feature $\vec{\boldsymbol{v}}_i^{(m)}$ as follows:

$$\boldsymbol{h}_i^{(m)} = \sigma_{\boldsymbol{h}}(\boldsymbol{h}_i^{(m)}, \boldsymbol{m}_i^{(m)}), \quad \vec{\boldsymbol{v}}_i^{(m)} = \vec{\boldsymbol{v}}_i^{(m)} + \vec{\boldsymbol{m}}_i^{(m)}. \tag{15}$$

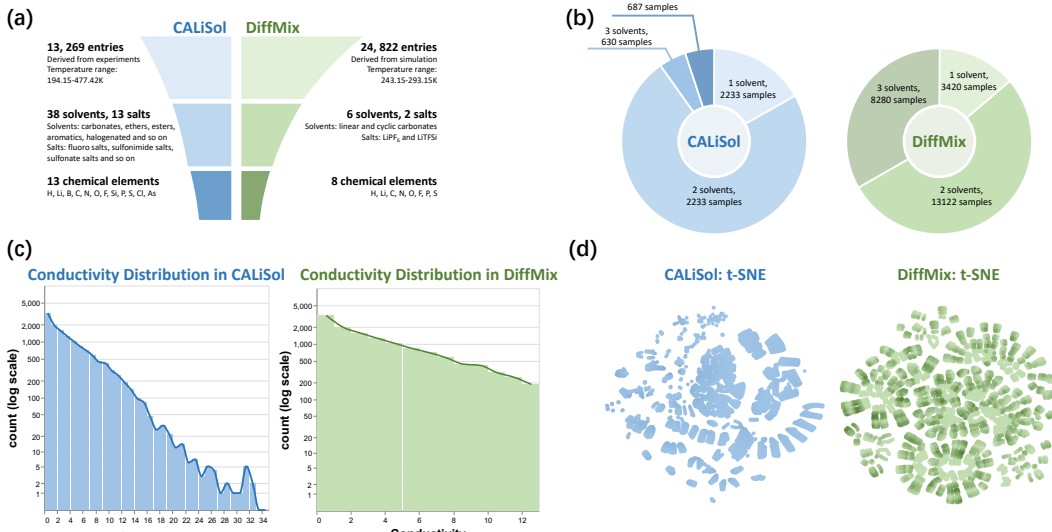

Figure 2: Overview of the CALiSol and DiffMix datasets. (a) Summary of dataset statistics. (b) Composition diversity in terms of the number of solvents used per sample. CALiSol exhibits a more diverse mixture design space, while DiffMix contains a greater number of samples with 2 or 3 solvents. (c) Log-scaled histograms of conductivity values indicate that CALiSol exhibits a broader range and a more pronounced long-tail distribution than DiffMix dataset. (d) The t-SNE visualization [60] reveals distinct structural characteristics in each dataset. CALiSol, derived from experimental measurements, exhibits greater diversity with many scattered outliers, reflecting heterogeneous sampling. In contrast, DiffMix shows a more uniform and clustered distribution, consistent with its simulation-based origin.

## 4 Experiments

In this section, we first introduce the two large-scale electrolyte datasets, CALiSol and DiffMix, covering their composition, property distributions, and geometric graph construction in § 4.1, and provide corresponding visualizations in Fig. 2. We then describe our experimental setup, including baseline models, GeoMix implementation, and evaluation metrics in § 4.2. The main results in § 4.3 highlight GeoMix's superior performance, and ablation studies in § 4.4 assess the impact of key components. Further details are provided in Appendix B.

### 4.1 Datasets

**CALiSol [19] and DiffMix [11] Datasets**     These two datasets provide comprehensive resources for electrolyte conductivity analysis. CALiSol contains 13,269 experimentally measured ionic conductivity entries (after filtering 556 incomplete points) spanning 13 lithium salts and 38 solvents across carbonate, ether, and ester classes, with broad coverage of salt concentrations and temperatures to reflect experimental diversity. In contrast, the simulation-driven DiffMix dataset employs systematic discretization with 24,822 formulations focusing on two salts (LiPF$_6$, LiTFSI) and up to three solvents selected from six carbonate types, quantizing salt concentrations into seven levels and normalizing solvent mass fractions via coarse sampling. Both datasets provide metadata including solvent proportions and measurement conditions, with full parameter ranges detailed in Appendix B.

**Geometric Graph Construction**     To facilitate geometric modeling, we further augmented the dataset with geometric molecular graph representations. We obtained molecular conformations from PubChem [61] when available. For molecules not found in PubChem, we generate 3D structures using RDKit [62] or download conformations from the Materials Project [63]. Each molecule is represented as a geometric graph, where atoms are nodes and edges are formed between atoms within a cutoff distance of 6 Å. Edge weights correspond to the Euclidean distances between atoms. Node features include atomic numbers and one-hot encodings of atom types.

**Data Splitting Strategy**     For both CALiSol and DiffMix datasets, we adopt a standard random split of 70%/20%/10% into train, validation, and test sets, respectively. Splitting is performed

Table 1: Results of MSE and Pearson correlation coefficient on CALiSol dataset and DiffMix dataset. Bold values indicate the best performance, while underlined values indicate the second-best.

| | Models | Symmetries | CALiSol | | DiffMix | |
|---|---|---|---|---|---|---|
| | | | MSE $\downarrow$ | Pearson $r \uparrow$ | MSE $\downarrow$ | Pearson $r \uparrow$ |
| MLP-based | MLP | None | 3.657 | 0.906 | 1.363 | 0.874 |
| | MM-MoLFormer [12] | | 5.488 | 0.825 | 1.901 | 0.812 |
| Topology-based | MolSets-Conv [13] | Permutation | 2.230 | 0.924 | 1.440 | 0.868 |
| | MolSets-SAGE [13] | | 2.751 | 0.909 | 0.708 | 0.937 |
| Geometry-based | EGNN-att [54] | Set-SE(3) | 2.666 | 0.908 | 0.752 | 0.930 |
| | TFN-att [55] | | 1.808 | 0.946 | 0.804 | 0.921 |
| | EGNN-linear [54] | | 1.461 | 0.951 | 0.195 | 0.988 |
| | TFN-linear [55] | | 1.107 | 0.967 | 0.285 | 0.973 |
| Geometry-based | GeoMix-EGNN | Set-SE(3) | 0.552 | 0.985 | 0.088 | 0.992 |
| | GeoMix-TFN | | **0.432** | **0.987** | **0.035** | **0.997** |

independently for CALiSol and DiffMix with a fixed seed to ensure reproducibility. No electrolyte formulation is shared across different subsets.

## 4.2 Experiment Setup

**Baselines** Previous studies such as MM-MoLFormer [12], MolSets [13], and Uni-ELF [18] have proposed individual models and conducted comparisons within limited baselines, but no prior work has provided a unified evaluation across modeling families using standardized datasets and metrics.

We compare with diverse baselines across architectural families: 1) **MLP-based**: We use a vanilla MLP taking the proportion vector as input, and MM-MoLFormer [12], which combines pretrained molecular embeddings with proportion and environment features. These models do not respect the symmetry constraints discussed in § 3.1 and exhibit limited generalization. 2) **Topology-based**: We adopt MolSets [13], a recent framework designed for molecular mixture modeling, and select GraphConv and GraphSAGE as representative backbones (denoted as MolSets-Conv and MolSets-SAGE). 3) **Geometry-based**: To account for spatial effects such as steric hindrance and solvation structure, we employ equivariant GNNs including EGNN [54] and TFN [55]. Each is paired with either a composition-weighted average (-linear) or a learnable attention-based aggregator (-att), yielding four geometric baselines: EGNN-att, TFN-att, EGNN-linear, and TFN-linear. The specific hyperparameters of the model are shown in Appendix B. Other recent geometry-based methods, such as DiffMix [11] and Uni-ELF [18], are not included in our baselines due to the lack of publicly available implementations.

**Implementation** For our GeoMix, we test it using two classical equivariant networks, EGNN and TFN, as the backbone for intramolecular embedding. Our method is consistent with the baseline in terms of hyperparameters such as the number of equivariant graph network layers and MLP layers. For other hyperparameters and other implementation details, please refer to Appendix B. We evaluate model performance using MSE and Pearson correlation coefficient ($r$) between the predicted and measured conductivity values.

## 4.3 Main Results

Table 1 summarizes the predictive performance of various baseline models and our proposed GeoMix on the CALiSol and DiffMix datasets. From these results, we observe the following: **1. Overall performance**: GeoMix consistently achieves the best performance across all metrics on both datasets, demonstrating its effectiveness in modeling molecular mixtures. **2. Advantage of geometry-based methods**: Geometry-based methods outperform both MLP-based and standard Topology-based baselines, highlighting the importance of incorporating molecular geometry into the learning process. **3. Benefit of higher-degree features**: Among geometric models, those using TFN as the backbone generally surpass those based on EGNN, suggesting that modeling higher-degree geometric features is particularly beneficial in mixture-related tasks. We further visualize the regression performance of some selected models in Fig. 3, which clearly illustrates the superiority of GeoMix-based methods in both accuracy and error consistency.

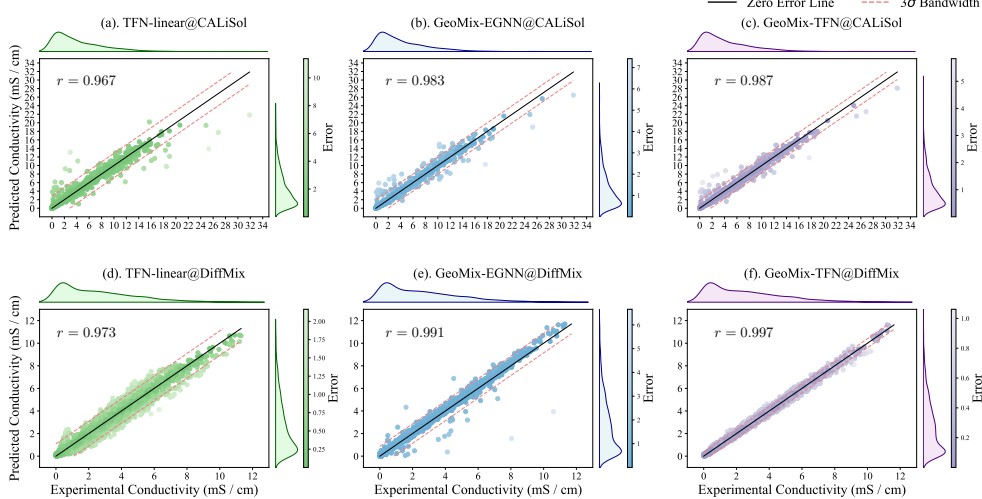

Figure 3: Regression plots for electrolyte conductivity prediction. (a–c) show results on the CALiSol dataset, using TFN-linear, GeoMix-EGNN, and GeoMix-TFN, respectively. (d–f) show corresponding results on the DiffMix dataset with the same model order.

## 4.4 Ablation Studies

To evaluate the individual contributions of our proposed components, we perform a series of ablation studies on the CALiSol dataset. All experiments are based on the GeoMix-EGNN model, which adopts the EGNN backbone due to its architectural simplicity and suitability for controlled analysis.

**Proportion Embedding** In the ablation variant denoted as "Multiply" in Table 2, we remove the proportion information from atomic features and instead apply scalar weighting to each per-molecule graph embedding before aggregation. This strategy mirrors traditional approaches that treat component ratios as external mixing coefficients rather than intrinsic modulators of molecular behavior. Table 2 reveals that embedding proportions into node features can effectively improve the effect of the model in modeling mixture systems.

**The Form of the Transformation Matrix** We conduct ablation studies to examine how different parameterizations of the transformation matrix $\boldsymbol{R}_{ij}^{(m,n)}$ affect model performance. We consider three alternative designs: 1)

Table 2: Ablations on CALiSol dataset.

| CALiSol | MSE ↓ | Pearson $r$ ↑ |
|---|---|---|
| GeoMix | 0.552 | 0.985 |
| *Proportion Embedding* | | |
| Multiply | 3.657 | 0.906 |
| *Transformation Matrix's Form* | | |
| Quaternion | 0.702 | 0.981 |
| 6D vector | 0.574 | 0.983 |
| Graph-wise | 0.662 | 0.980 |
| *Linear v.s. Attention* | | |
| GeoMix-linear | 0.851 | 0.975 |
| *Noisy Nodes Loss* | | |
| *w/o* Noisy Nodes | 1.213 | 0.969 |

**Quaternion**: The rotation matrix is derived from a learned unit quaternion, ensuring orthogonality with $\det(\boldsymbol{R}_{ij}^{(m,n)}) = 1$ (*i.e.*, rigid transformation without scaling). 2) **6D Vector**: The rotation is parameterized by two learned vectors followed by Gram-Schmidt orthogonalization, providing a continuous and differentiable representation of SO(3) [59], also enforcing $\det(\boldsymbol{R}_{ij}^{(m,n)}) = 1$. 3) **Graph-wise**: Instead of learning node-pair-specific transformations, we learn a single transformation per graph pair by pooling node features. This enforces graph-level rigidity and ignores local structural flexibility. From the results in Table 2, we observe that: (1) Models with non-rigid, node-wise transformations (GeoMix, Quaternion, 6D vector) outperform the graph-rigid variant (Graph-wise), suggesting the importance of fine-grained local transformations. (2) Among non-rigid variants, GeoMix, which does not constrain the transformation matrix to be orthogonal or have unit determinant, achieves the best performance. This indicates that allowing scaling transformations further enhances flexibility and expressivity in modeling cross-graph geometry.

**Graph-Level Readout** As shown in Table 1, the simple linear average readout performs competitively among baseline models. To investigate its effect within our framework, we replace the original graph-level readout in GeoMix with the linear average strategy, resulting in an ablation

variant denoted as GeoMix-linear. As reported in Table 2, GeoMix-linear yields lower performance compared to the original GeoMix. This suggests that the full model benefits from the rich, attention-based aggregation mechanism, which we attribute to the GIN. The learned transformation-aware interactions provide expressive relational cues between molecules, enabling the self-attention module to generate more informative graph-level embeddings beyond simple averaging.

**Noisy Nodes Loss** We further evaluate the impact of the Noisy Nodes loss [57] by removing it from the training objective, denoted as "*w/o* Noisy Nodes" in Table 2. The results demonstrate that incorporating this regularization consistently improves the model's performance. We attribute this to its role in enhancing representation robustness by perturbing node features during training.

## 5 Conclusion

We presented GeoMix, a Set-SE(3) equivariant framework for predicting properties of molecular mixtures. GeoMix leverages equivariant graph neural networks to embed individual species and introduces a Geometric Interaction Network (GIN) to perform intermolecular message passing via pre-constructing local frames. To support systematic evaluation, we curated two large-scale datasets with geometric features, CALiSol and DiffMix, focused on electrolyte conductivity. Based on these datasets, we established a unified benchmark for electrolyte conductivity prediction. Experimental results demonstrate that GeoMix significantly outperforms existing baselines, highlighting the importance of geometry-aware modeling in mixture systems. Future work may extend this framework to other multi-component tasks in chemistry and materials science.

## 6 Limitations

Despite the promising results, this work has several limitations that warrant further investigation. First, the available datasets remain limited in both scope and diversity. In particular, no datasets currently provide molecular dynamics-based microscopic structural information, and there is a lack of conductivity datasets covering a broader range of chemistries, such as sodium-ion or potassium-ion electrolytes. Second, the encoder architecture adopted in GeoMix is relatively simple, leaving room for improvement.

In future work, we plan to incorporate richer descriptors and leverage recent advances in graph neural networks to enhance both the accuracy and the generalization ability of the model [64; 65]. We also aim to curate and release new benchmark datasets that include microscopic structural features and broader compositional coverage, thereby enabling more comprehensive evaluation and fostering further research in mixture property prediction.

## Acknowledgement

This work was jointly supported by the following projects: the National Natural Science Foundation of China (No. 62376276); Beijing Nova Program (No. 20230484278); the Fundamental Research Funds for the Central Universities; the Research Funds of Renmin University of China (23XNKJ19); Public Computing Cloud, Renmin University of China.

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

# Contents of Appendix

# A   Theoretical Details

## A.1   Expressivity of Intermolecular Transformation Matrix

**Lemma A.1.** *For any* $\mathrm{O}(n)$*-equivariant function* $\hat{f}(\vec{\boldsymbol{Z}})$*, it must fall into the subspace spanned by the columns of* $\vec{\boldsymbol{Z}}$*, namely, there exists a function* $s(\vec{\boldsymbol{Z}})$*, satisfying* $\hat{f}(\vec{\boldsymbol{Z}}) = \vec{\boldsymbol{Z}}s(\vec{\boldsymbol{Z}})$*.*

*Proof.* The proof is given by [66]. Essentially, suppose $\vec{\boldsymbol{Z}}^{\perp}$ is the orthogonal complement of the column space of $\vec{\boldsymbol{Z}}$. Then there must exit functions $s(\vec{\boldsymbol{Z}})$ and $s^{\perp}(\vec{\boldsymbol{Z}})$, satisfying $\hat{f}(\vec{\boldsymbol{Z}}) = \vec{\boldsymbol{Z}}s(\vec{\boldsymbol{Z}}) + \vec{\boldsymbol{Z}}^{\perp}s^{\perp}(\vec{\boldsymbol{Z}})$. We can always find an orthogonal transformation $\boldsymbol{O}$ allowing $\boldsymbol{O}\vec{\boldsymbol{Z}} = \vec{\boldsymbol{Z}}$ while $\boldsymbol{O}\vec{\boldsymbol{Z}}^{\perp} = -\vec{\boldsymbol{Z}}^{\perp}$. With this transformation $\boldsymbol{O}$, we have $\hat{f}(\boldsymbol{O}\vec{\boldsymbol{Z}}) = \hat{f}(\vec{\boldsymbol{Z}}) = \vec{\boldsymbol{Z}}s(\vec{\boldsymbol{Z}}) + \vec{\boldsymbol{Z}}^{\perp}s^{\perp}(\vec{\boldsymbol{Z}})$, and $\boldsymbol{O}\hat{f}(\vec{\boldsymbol{Z}}) = \vec{\boldsymbol{Z}}s(\vec{\boldsymbol{Z}}) - \vec{\boldsymbol{Z}}^{\perp}s^{\perp}(\vec{\boldsymbol{Z}})$. The equivariance property of $\hat{f}$ implies $\vec{\boldsymbol{Z}}s(\vec{\boldsymbol{Z}}) + \vec{\boldsymbol{Z}}^{\perp}s^{\perp}(\vec{\boldsymbol{Z}}) = \vec{\boldsymbol{Z}}s(\vec{\boldsymbol{Z}}) - \vec{\boldsymbol{Z}}^{\perp}s^{\perp}(\vec{\boldsymbol{Z}})$, which derives $s^{\perp}(\vec{\boldsymbol{Z}}) = 0$. Hence, the proof is concluded. □

**Lemma A.2.** *If the* $\mathrm{O}(n)$*-equivariant function* $\hat{f}(\vec{\boldsymbol{Z}})$ *lies in the subspace spanned by the columns of* $\vec{\boldsymbol{Z}}$*, then there exists a function* $\sigma$ *satisfying* $\hat{f}(\vec{\boldsymbol{Z}}) = \vec{\boldsymbol{Z}}\sigma(\vec{\boldsymbol{Z}}^{\top}\boldsymbol{Z})$*.*

*Proof.* The proof is provided by Corollary 2 in [67]. The basic idea is that $\hat{f}(\vec{\boldsymbol{Z}})$ can be transformed to $\hat{f}(\vec{\boldsymbol{Z}}) = \vec{\boldsymbol{Z}}\eta(\vec{\boldsymbol{Z}})$ where $\eta(\vec{\boldsymbol{Z}})$ is $\mathrm{O}(n)$-invariant. According to Lemma 1-2 in [67], $\eta(\vec{\boldsymbol{Z}})$ must be written as $\eta(\vec{\boldsymbol{Z}}) = \sigma(\vec{\boldsymbol{Z}}^{\top}\vec{\boldsymbol{Z}})$, which completes the proof. □

**Theorem 3.1** (Expressivity of Intermolecular Transformation Matrix). *Given geometric graphs* $\mathcal{G}_m$ *and* $\mathcal{G}_n$*, with* $\vec{\boldsymbol{F}}_m, \vec{\boldsymbol{F}}_n$ *being respective frames, any matrix* $\overleftrightarrow{\boldsymbol{I}}_{mn}$ *satisfying* $\mathrm{SO}(3)$*-equivariance* $\overleftrightarrow{\boldsymbol{I}}_{mn} \xmapsto{\boldsymbol{R}_m, \boldsymbol{R}_n \in \mathrm{SO}(3)} \boldsymbol{R}_m \overleftrightarrow{\boldsymbol{I}}_{mn} \boldsymbol{R}_n^{\top}$ *must take the form like Eq. (10) as* $\overleftrightarrow{\boldsymbol{I}}_{mn}(\vec{\boldsymbol{F}}_m, \vec{\boldsymbol{F}}_n) = \vec{\boldsymbol{F}}_m \boldsymbol{R} \vec{\boldsymbol{F}}_n^{\top}$*.*

*Proof.* First, we want to show that, $\overleftrightarrow{\boldsymbol{I}}_{mn}(\vec{\boldsymbol{F}}_m, \vec{\boldsymbol{F}}_n))$ is equivalent to $\overleftrightarrow{\boldsymbol{I}}_{mn}(\vec{\boldsymbol{F}}_m \otimes \vec{\boldsymbol{F}}_n))$. We know the Kronecker product $\vec{\boldsymbol{F}}_m \otimes \vec{\boldsymbol{F}}_n$ contains $9 \times 9$ elements, which could be divided into 9 small matrices of $3 \times 3$, each corresponding to the product of a row in $\vec{\boldsymbol{F}}_m = [\vec{\boldsymbol{u}}_1; \vec{\boldsymbol{u}}_2; \vec{\boldsymbol{u}}_3]$ and a column in $\vec{\boldsymbol{F}}_n = [\vec{\boldsymbol{v}}_1^{\top}; \vec{\boldsymbol{v}}_2^{\top}; \vec{\boldsymbol{v}}_3^{\top}]$, *i.e.* $\vec{\boldsymbol{F}}_m \otimes \vec{\boldsymbol{F}}_n = [\vec{\boldsymbol{u}}_i \otimes \vec{\boldsymbol{v}}_j^{\top}]$. Note $\vec{\boldsymbol{u}}_i \otimes \vec{\boldsymbol{v}}_j^{\top}$ is a rank-one matrix and $\vec{\boldsymbol{u}}_i, \vec{\boldsymbol{v}}_j$ are both unit vectors. Therefore, $\vec{\boldsymbol{u}}_i \otimes \vec{\boldsymbol{v}}_j^{\top}$ corresponds to two possibilities: $(\vec{\boldsymbol{u}}_i, \vec{\boldsymbol{v}}_j^{\top})$ or $(-\vec{\boldsymbol{u}}_i, -\vec{\boldsymbol{v}}_j^{\top})$. By analyzing the 9 submatrices, we can find that $\vec{\boldsymbol{F}}_m \otimes \vec{\boldsymbol{F}}_n$ also corresponds to two possible decomposition, namely $(\vec{\boldsymbol{F}}_m, \vec{\boldsymbol{F}}_n)$ and $(-\vec{\boldsymbol{F}}_m, -\vec{\boldsymbol{F}}_n)$. Note that during the construction, we agreed that $\det(\vec{\boldsymbol{F}}_m) = \det(\vec{\boldsymbol{F}}_n) = 1$, so there is a direct bijection between $\vec{\boldsymbol{F}}_m \otimes \vec{\boldsymbol{F}}_n$ and $(\vec{\boldsymbol{F}}_m, \vec{\boldsymbol{F}}_n)$, which means $\overleftrightarrow{\boldsymbol{I}}_{mn}(\vec{\boldsymbol{F}}_m, \vec{\boldsymbol{F}}_n))$ is equivalent to $\overleftrightarrow{\boldsymbol{I}}_{mn}(\vec{\boldsymbol{F}}_m \otimes \vec{\boldsymbol{F}}_n))$.

Then, we use the vectorization operator $\mathrm{vec}(\cdot)$ to prove $\overleftrightarrow{\boldsymbol{I}}_{mn}(\vec{\boldsymbol{F}}_m \otimes \vec{\boldsymbol{F}}_n))$ must take the form:

$$\mathrm{vec}(\overleftrightarrow{\boldsymbol{I}}_{mn}(\vec{\boldsymbol{F}}_m \otimes \vec{\boldsymbol{F}}_n)) = \mathrm{vec}(\vec{\boldsymbol{F}}_m \boldsymbol{R} \vec{\boldsymbol{F}}_n^{\top}) = (\vec{\boldsymbol{F}}_n \otimes \vec{\boldsymbol{F}}_m)\,\mathrm{vec}(\boldsymbol{R}). \tag{16}$$

Note that $\vec{\boldsymbol{F}}_n, \vec{\boldsymbol{F}}_m$ are constructed frames, which is full-rank. Then we find $\vec{\boldsymbol{F}}_n \otimes \vec{\boldsymbol{F}}_m$ will be also full-rank, spanning the whole space $\mathbb{R}^9$ where $\mathrm{vec}(\overleftrightarrow{\boldsymbol{I}}_{mn}(\vec{\boldsymbol{F}}_m, \vec{\boldsymbol{F}}_n)$ lies in. According to Lemma A.2, we know $\mathrm{vec}(\overleftrightarrow{\boldsymbol{I}}_{mn})$ must satisfy:

$$\mathrm{vec}(\overleftrightarrow{\boldsymbol{I}}_{mn}) = (\vec{\boldsymbol{F}}_n \otimes \vec{\boldsymbol{F}}_m)\sigma((\vec{\boldsymbol{F}}_n \otimes \vec{\boldsymbol{F}}_m)^{\top}(\vec{\boldsymbol{F}}_n \otimes \vec{\boldsymbol{F}}_m)). \tag{17}$$

Moreover, we have such formula:

$$(\vec{\boldsymbol{F}}_n \otimes \vec{\boldsymbol{F}}_m)^{\top}(\vec{\boldsymbol{F}}_n \otimes \vec{\boldsymbol{F}}_m) = (\vec{\boldsymbol{F}}_n^{\top}\vec{\boldsymbol{F}}_n) \otimes (\vec{\boldsymbol{F}}_m^{\top}\vec{\boldsymbol{F}}_m) = \boldsymbol{I}_{3\times3} \otimes \boldsymbol{I}_{3\times3} = \boldsymbol{I}_{9\times9}, \tag{18}$$

which is a constant matrix. Thus $\sigma((\vec{\boldsymbol{F}}_n \otimes \vec{\boldsymbol{F}}_m)^{\top}(\vec{\boldsymbol{F}}_n \otimes \vec{\boldsymbol{F}}_m)) = \sigma(\boldsymbol{I}_{9\times9})$ will be learnable parameters, which are equivalent to the $\mathrm{vec}(\boldsymbol{R})$ here. Up to this point, the matrix $\overleftrightarrow{\boldsymbol{I}}_{mn}$ appears to exhibit $\mathrm{O}(3)$-equivariance. However, it is crucial to observe that the frame constructions $\mathcal{F}_n$ and $\mathcal{F}_m$ are only $\mathrm{SO}(3)$-equivariant. Consequently, the overall transformation reduces to $\mathrm{SO}(3)$-equivariance. This completes the proof. □

## A.2 Equivariance/Invariance of GeoMix

### A.2.1 Enhancement of Equivariance/Invariance

In order to prove the equivariance/invariance of the entire GeoMix model, we need to strengthen the conditions.

**Theorem A.3** (Enhancement of Equivariance/Invariance). *If each module of GeoMix is equivariant/invariant, then the entire GeoMix will also be equivariant/invariant.*

*Proof.* Consider a sequence composed of functions $\{\phi_i : \mathcal{X}^{(i-1)} \to \mathcal{X}^{(i)}\}_{i=1}^N$ equivariant to a same group $G$, the equivariance lead to an interesting property that

$$\phi_N \circ \cdots \circ \phi_{i+1} \circ \rho_{\mathcal{X}^{(i)}}(g)\phi_i \circ \cdots \circ \phi_1 = \phi_N \circ \cdots \circ \phi_{j+1} \circ \rho_{\mathcal{X}^{(j)}}(g)\phi_j \circ \cdots \circ \phi_1,$$

holds for all $i, j = 1, 2, \ldots, N$ and $g \in G$, which means that the group elements $g$ can be freely exchanged in the composite sequence of equivariant functions. In particular, if one of the equivariant functions (*e.g.* $\phi_k$) is replaced by an invariant function, the group element $g$ will be absorbed, that means

$$\phi_N \circ \cdots \circ \phi_k \circ \cdots \circ \phi_{i+1} \circ \rho_{\mathcal{X}^{(i)}}(g)\phi_i \circ \cdots \circ \phi_1 = \phi_N \circ \cdots \circ \phi_1.$$

holds for all $g \in G$ but only $i = 1, 2, \ldots, k$. Although $\phi_N \circ \cdots \circ \phi_k$ is still equivariant, because the group elements must be input starting from $\phi_1$, the overall $\phi_N \circ \cdots \circ \phi_1$ is still an invariant function. □

Specifically, we need to prove the following:

1. SO(3)-equivariance of local frame.
2. SO(3)-equivariance of learnable transformations.

### A.2.2 Equivariance of Local Frame

**Theorem A.4** (Equivariance of Local Frame). *The construction of local frame is* SO(3)*-equivariance.*

*Proof.* We prove that the local frame $\vec{F}_m$ is SO(3)-equivariant with respect to the target graph $\mathcal{G}_m$. More formally, for any orthogonal matrix $Q \in \mathbb{R}^{3 \times 3}$, the local frame should satisfy:

$$Q\vec{F}_m = \texttt{LocalFrame}(Q\vec{X}_m). \tag{19}$$

Formally, we have

$$\begin{aligned}
\mathrm{Cov}(Q\vec{X}_m) &= \frac{1}{N_m}Q\vec{X}_m(Q\vec{X}_m)^\top = Q\frac{1}{N_m}\vec{X}_m\vec{X}_m^\top Q^\top \\
&= Q\mathrm{Cov}(\vec{X}_m)Q^\top = Q\vec{F}_m\Lambda_m\vec{F}_m^\top Q^\top \\
&= (Q\vec{F}_m)\Lambda_m(Q\vec{F}_m)^\top.
\end{aligned} \tag{20}$$

Concluding we showed that an SO(3) transformation on the original coordinate matrix $\vec{X}_m$ results in the same transformation on the output local frame $\vec{F}_m$. □

### A.2.3 Equivariance of Learnable Transformations

**Theorem A.5** (Equivariance of Learnable Transformations). *The construction of learnable transformations is* SO(3)*-equivariance.*

*Proof.* We prove the message $\vec{m}_{ij}^{(m)}$ and $\vec{m}_{ij}^{(m,n)}$ in Eq. (13) is SO(3) equivariant on the target graph $\mathcal{G}_m$ and the source graph $\mathcal{G}_n$ respectively, that is, for any orthogonal matrix $Q_m, Q_n \in \mathbb{R}^{3 \times 3}$, the message should satisfy:

$$\begin{aligned}
Q_n\vec{m}_{ij}^{(m,n)} &= \texttt{MessageSource}(Q_n\vec{X}_n, Q_n\vec{V}_n), \\
Q_m\vec{m}_{ij}^{(m)} &= \texttt{MessageTarget}(Q_m\vec{X}_m).
\end{aligned} \tag{21}$$

It is easy to see $\boldsymbol{m}_{ij}^{(m,n)}$ is invariant on both the source and the target graph. So,

$$
\begin{aligned}
&\sigma_{\vec{\boldsymbol{x}}}(\boldsymbol{m}_{ij}^{(m,n)})\boldsymbol{Q}_n\vec{\boldsymbol{x}}_j^{(n)} + \sigma_{\vec{\boldsymbol{v}}}(\boldsymbol{m}_{ij}^{(m,n)})\boldsymbol{Q}_n\vec{\boldsymbol{v}}_j^{(n)} \\
=&\boldsymbol{Q}_n\sigma_{\vec{\boldsymbol{x}}}(\boldsymbol{m}_{ij}^{(m,n)})\vec{\boldsymbol{x}}_j^{(n)} + \boldsymbol{Q}_n\sigma_{\vec{\boldsymbol{v}}}(\boldsymbol{m}_{ij}^{(m,n)})\vec{\boldsymbol{v}}_j^{(n)} \\
=&\boldsymbol{Q}_n\vec{\boldsymbol{m}}_{ij}^{(m,n)},
\end{aligned}
\tag{22}
$$

which is SO(3) equivariant on the source graph.

We have proven that the local frame $\vec{\boldsymbol{F}}_m$ is SO(3) equivariant on the target graph, so $\vec{\boldsymbol{F}}_n$ is SO(3) equivariant on the source graph. We have:

$$
\begin{aligned}
&\boldsymbol{Q}_m\vec{\boldsymbol{F}}_m\boldsymbol{R}_{ij}^{(m,n)}(\boldsymbol{Q}_n\vec{\boldsymbol{F}}_n)^\top\boldsymbol{Q}_n\vec{\boldsymbol{m}}_{ij}^{(m,n)} + \boldsymbol{Q}_m\vec{\boldsymbol{F}}_m\boldsymbol{t}_{ij}^{(m,n)} \\
=&\boldsymbol{Q}_m\vec{\boldsymbol{F}}_m\boldsymbol{R}_{ij}^{(m,n)}\vec{\boldsymbol{F}}_n^\top\boldsymbol{Q}_n^\top\boldsymbol{Q}_n\vec{\boldsymbol{m}}_{ij}^{(m,n)} + \boldsymbol{Q}_m\vec{\boldsymbol{t}}_{ij}^{(m)} \\
=&\boldsymbol{Q}_m\overleftrightarrow{\boldsymbol{I}}_{ij}^{(m,n)}\vec{\boldsymbol{m}}_{ij}^{(m,n)} + \boldsymbol{Q}_m\vec{\boldsymbol{t}}_{ij}^{(m)} \\
=&\boldsymbol{Q}_m\vec{\boldsymbol{m}}_{ij}^{(m)},
\end{aligned}
\tag{23}
$$

which is SO(3) equivariant on the target graph. $\square$

# B  More Experiment Details and Results

Our code and dataset are available at https://github.com/GLAD-RUC/GeoMix.

## B.1  Dataset Details

In this part, we provide supplementary details and visualizations of the datasets introduced in the main text. These include the distribution of molecular species, property ranges, and composition coverage, as illustrated in Table 3.

Table 3: Summary statistics of the CALiSol [19] and DiffMix [11] datasets, including the number of samples, solvent and salt types, elemental coverage, maximum number of solvents per formulation, solute concentration ranges, and temperature ranges.

|  | CALiSol | DiffMix |
| --- | --- | --- |
| Number of samples | 13,269 | 24,822 |
| Number of unique solvents | 38 | 6 |
| Solvent list | EC, PC, DMC, DEC, DME, DMSO, AN, MOEMC, TFP, EA, MA, FEC, DOL, 2-MeTHF, DMM, Freon 11, Methylene chloride, THF, Toluene, Sulfolane, 2-Glyme, 3-Glyme, 4-Glyme, 3-Me-2-Oxazolidinone, 3-MeSulfolane, Ethyldiglyme, DMF, Ethylbenzene, Ethylmonoglyme, Benzene, g-Butyrolactone, Cumene, Propylsulfone, Pseudocumeme, TEOS, m-Xylene, o-Xylene | EC, PC, FEC, EMC, DEC, DMC |
| Number of unique salts | 13 | 2 |
| Salt list | $LiPF_6$, $LiBF_4$, LiFSI, LiTDI, LiPDI, LiTFSI, $LiClO_4$, $LiAsF_6$, LiBOB, $LiCF_3SO_3$, LiBPFPB, LiBMB, $LiN(CF_3SO_2)_2$ | $LiPF_6$, LiTFSI |
| Elements covered | H, Li, B, C, N, O, F, Si, P, S, Cl, As | H, Li, C, N, O, F, P, S |
| Maximum number of solvents per sample | 4 | 3 |
| Solute composition range | 0.0286-2.37 mol/kg and 0.0771-4.00 mol/L | 0.025-3.0 mol/kg [4] |
| Temperature range | 194.15-477.42K | 243.15K-293.15K |

## B.2 Implementation Details

In this part, we describe the implementation details. Each model architecture remains the same in both datasets.

- **MLP**: A 3-layer multilayer perceptron is used, with each hidden layer having 64 dimensions.
- **MM-MoLFormer**: We follow the settings in [12]. MM-MolFormer uses a pre-trained MoL-Former model [68], obtained from https://ibm.ent.box.com/v/MoLFormer-data. The molecular embeddings are concatenated with the corresponding proportion and temperature features, and then passed through a 2-layer MLP with hidden size 64 to generate the final prediction.
- **MolSets**: For both MolSets-Conv and MolSets-SAGE, we retain the original hyperparameter settings from [13], except that the hidden dimension of each graph neural network layer is increased to 64.
- **EGNN-att**: This variant employs a 4-layer EGNN backbone with 64 hidden dimensions per layer. The attention module takes 8 molecular embeddings as input, with 4 attention heads and 3 stacked attention layers. The resulting representation is concatenated with temperature $T$ and salt concentration $c$, and then passed through a 3-layer MLP for prediction.
- **TFN-att**: TFN-att shares the same configuration as EGNN-att except for the backbone, which is replaced with a TFN. The TFN backbone uses a maximum angular momentum quantum number of $\max L = 2$, and each irreducible representation (irrep) has 8 channels.
- **EGNN-linear**: This model uses the same EGNN backbone as EGNN-att. Instead of an attention module, a simple non-learnable linear average (scatter) operation is applied to obtain the mixture representation
- **TFN-linear**: This model adopts the same TFN backbone as TFN-att, and replaces the attention module with a non-learnable linear average operation, as in EGNN-linear.
- **GeoMix-EGNN**: GeoMix-EGNN adopts a multi-channel EGNN as the backbone, with 8 equivariant channels. Other hyperparameters follow the EGNN-att setting. The GIN consists of 3 layers. The Noisy Nodes loss [57] is applied with hyperparameter $\gamma = 128$.
- **GeoMix-TFN**: GeoMix-TFN uses the same TFN backbone as TFN-att. The GIN also consists of 3 layers. The Noisy Nodes loss is applied with $\gamma = 128$, as in GeoMix-EGNN.

## B.3 Training Settings

All models are trained using the Adam optimizer with a learning rate of $5 \times 10^{-5}$, weight decay of $1 \times 10^{-12}$, and a maximum of 500 training epochs. The default batch size is 1024, except for GeoMix models, which use a reduced batch size of 128 due to GPU memory constraints. We fix the random seed to 7 for all experiments to ensure reproducibility.

We run our methods mainly on A100 80GB GPUs.

## B.4 OOD Evaluation across Conductivity.

In practical chemical research, it is often necessary to identify substances with high ionic conductivity based on prior knowledge of low-conductivity samples, which naturally poses an out-of-distribution (OOD) generalization challenge. However, most existing machine learning models struggle to generalize reliably under such distribution shifts. Here, we specifically consider an OOD split where low-conductivity samples are used for training and validation, while high-conductivity samples are reserved for testing.

**Experiment Setup.** To evaluate our model's robustness in this setting, we selected samples with conductivity $\leq 10$ mS/cm as the training and validation sets, and those with conductivity $> 10$ mS/cm as the test set on the CALiSol dataset. The training and validation sets were split randomly in a 4:1 ratio. The high-conductivity test set contains 1,244 samples, ensuring that this split is statistically meaningful. Here, we evaluate only EGNN-linear and TFN-linear, the two best-performing models

---

[4]The salt concentrations are discretized into seven levels: {0.025, 0.5, 1.0, 1.5, 2.0, 2.5, 3.0} molal. For each formulation, the solvent mass fractions vary from 0 to 1 in increments of 0.2 and are normalized to sum to one.

Table 4: Results of OOD evaluation across conductivity on CALiSol dataset. Bold values indicate the best performance, while underlined values indicate the second-best.

| Models | MSE ↓ | MAE ↓ | Pearson $r$ ↑ | Spearman $r$ ↑ |
|---|---|---|---|---|
| EGNN-linear [54] | 32.720 | 4.457 | 0.253 | 0.341 |
| TFN-linear [55] | 24.218 | 3.585 | 0.397 | 0.508 |
| GeoMix-EGNN | **17.132** | **2.853** | **0.579** | 0.571 |
| GeoMix-TFN | 19.427 | 2.925 | 0.436 | **0.609** |

in § 4.2, as baselines. During training, we observed that attention-based aggregation of graph-level information tended to overfit. To mitigate this, we applied a dropout rate of 0.1 to the attention layer and set the attention temperature to 2.0. All other settings remained unchanged.

**Results.** The results for this OOD split based on conductivity values are summarized in Table 4. Both baseline models, EGNN-linear and TFN-linear, exhibit limited generalization. Our proposed models, GeoMix-EGNN and GeoMix-TFN, show significantly lower MSE and higher correlation metrics compared to the baselines, demonstrating their effectiveness in capturing trends in this OOD setting. These results highlight that our approach improves predictive performance while enhancing OOD generalization under conductivity-based distribution shifts.

## B.5   OOD Evaluation across Temperature.

Because the free diffusion rate of most materials at ambient temperature is low, it is common to perform molecular dynamics simulations at elevated temperatures and then extrapolate to lower temperatures using the Arrhenius empirical relation [8]. Therefore, we explored the consistency of model predictions with respect to temperature.

**Experiment Setup.** To empirically assess this, we performed an OOD generalization experiment where we trained the model on data with temperature $\leq 320$ K (using a train/valid split of 4:1), and evaluated on samples with temperature $> 320$ K. All other settings remained unchanged.

**Results.** Results in Table 5 show that GeoMix achieves significantly better predictive performance on this challenging temperature extrapolation task compared to other baseline models, suggesting that the learned representations effectively capture physically meaningful temperature trends without requiring hard-coded constraints.

Table 5: Results of OOD evaluation across temperature on CALiSol dataset. Bold values indicate the best performance, while underlined values indicate the second-best.

| Models | MSE ↓ | MAE ↓ | Pearson $r$ ↑ | Spearman $r$ ↑ |
|---|---|---|---|---|
| MLP | 32.432 | 3.857 | 0.432 | 0.556 |
| MM-MoLFormer [12] | 22.935 | 2.837 | 0.512 | 0.560 |
| MolSets-Conv [13] | 9.930 | 2.196 | 0.839 | 0.845 |
| MolSets-SAGE [13] | 9.193 | 2.135 | 0.784 | 0.802 |
| EGNN-att [54] | 8.134 | 1.594 | 0.813 | 0.861 |
| TFN-att [55] | 6.516 | 1.383 | 0.853 | 0.887 |
| EGNN-linear [54] | 7.675 | 1.616 | 0.827 | 0.847 |
| TFN-linear [55] | 4.780 | 1.286 | 0.914 | 0.930 |
| GeoMix-EGNN | 2.366 | **0.883** | 0.950 | **0.948** |
| GeoMix-TFN | **2.354** | 0.917 | **0.952** | 0.945 |

## B.6   Experiment of coordinate perturbation.

For complex molecular systems, the initial conformations used in modeling may not always be fully accurate. To assess the robustness of our model to perturbations in the input geometries, we conducted an additional robustness evaluation.

**Experiment Setup.** All other settings remained unchanged, except that Gaussian noise with a standard deviation of $\sigma = 0.3$ Å was added to the atomic coordinates in the test set to simulate conformational uncertainty.

**Results.** As shown in Table 6, most models—including GeoMix-EGNN—exhibit only a marginal performance drop, indicating stability under moderate geometric perturbations. Notably, models based on the TFN encoder show a much larger performance drop, which may stem from their reliance on higher-degree equivariant features (e.g., spherical harmonics) that could be sensitive to atomic coordinate perturbations.

Table 6: Results of coordinate perturbation on CALiSol dataset. Bold values indicate the best performance.

| Models | MSE ↓ | MAE ↓ | Pearson $r$ ↑ | Spearman $r$ ↑ |
|---|---|---|---|---|
| MolSets-Conv [13] | 1.326 | 0.703 | 0.960 | 0.965 |
| MolSets-SAGE [13] | 1.748 | 0.794 | 0.946 | 0.958 |
| EGNN-att [54] | 1.919 | 0.587 | 0.942 | 0.965 |
| TFN-att [55] | 2.294 | 0.760 | 0.929 | 0.954 |
| EGNN-linear [54] | 1.114 | 0.555 | 0.967 | 0.976 |
| TFN-linear [55] | 2.393 | 0.937 | 0.929 | 0.944 |
| GeoMix-EGNN | **0.631** | **0.388** | **0.982** | **0.983** |
| GeoMix-TFN | 4.517 | 1.324 | 0.859 | 0.886 |

## B.7 Runtime Analysis on CALiSol dataset.

Results in Table 7 shows the runtime analysis comparing our methods with other main baselines on the CALiSol dataset.

Table 7: Runtime Analysis on CALiSol dataset.

| Models | Inference Time ($\times 10^{-2}$ s) | Memory Usage (GB) | MSE ↓ | Pearson $r$ ↑ |
|---|---|---|---|---|
| EGNN-linear [54] | 4.41 | 2.20 | 1.461 | 0.951 |
| TFN-linear [55] | 24.13 | 5.91 | 1.107 | 0.967 |
| EGNN-att [54] | 23.40 | 2.22 | 2.666 | 0.908 |
| TFN-att [55] | 43.20 | 5.91 | 1.808 | 0.946 |
| GeoMix-EGNN | 34.04 | 25.15 | 0.552 | 0.985 |
| GeoMix-TFN | 53.64 | 45.18 | 0.432 | 0.987 |

## B.8 Evaluation of Different Frame Construction Methods

As described in § 3.2, the local frame $\vec{F}_m$ for each molecule is constructed via eigen-decomposition of the covariance matrix $\mathrm{Cov}(\vec{X}_m)$. However, the axis orientation of $\vec{F}_m$ obtained from standard PCA is not unique, as simultaneous sign flips of eigenvectors can lead to instability in local frame alignment. To alleviate this ambiguity, we additionally test a deterministic version of PCA by enumerating all eight possible axis-orientation combinations of the eigenbasis and selecting the one with the largest coordinate sum, i.e.,

$$\vec{F}_m^* = \operatorname*{argmax}_{\vec{F}_m' \in \mathbb{F}(\vec{F}_m)} \mathbf{1}^\top \vec{F}_m' \mathbf{1}, \tag{24}$$

where $\mathbb{F}(\vec{F}_m)$ denotes the set of all sign permutations of the three principal axes. This procedure ensures a consistent frame orientation across molecules.

**Performance Comparison.** Table 8 summarizes the predictive performance of different frame construction methods under the same experimental setup as described in § 4.2. The results show that the deterministic (strict) PCA achieves performance comparable to the standard PCA, though with slightly higher errors in some cases. According to [56], the standard PCA-based frame construction, while not strictly equivariant, remains reliable and effective for defining local coordinate systems in

molecular modeling. The reduced flexibility of the strictly equivariant variant may account for its marginally lower performance. In addition, the methods in [16; 17] can also be used as alternatives, which we will study in future work.

Table 8: Results on the CALiSol dataset. *FC Methods* denotes the frame construction methods. The *strict PCA* refers to the deterministic variant defined in Eq. (24).

| Models | FC Methods | MSE $\downarrow$ | Pearson $r$ $\uparrow$ |
|---|---|---|---|
| GeoMix-EGNN | PCA | 0.552 | 0.985 |
| GeoMix-TFN | PCA | 0.432 | 0.987 |
| GeoMix-EGNN | strict PCA | 0.628 | 0.981 |
| GeoMix-TFN | strict PCA | 0.460 | 0.986 |

