# OpenReview forum: "Geometric Mixture Models for Electrolyte Conductivity Prediction"
_NeurIPS.cc/2025/Conference — NeurIPS 2025 poster_

### Official Review · Reviewer_Syf6 · 2025-06-17

**Clarity:** 2
**Significance:** 3
**Originality:** 2
**Rating:** 4
**Confidence:** 4

**Summary:**

The paper proposes GeoMix, a novel geometry-aware framework that preserves Set-SE(3) equivariance for accurate prediction of ionic conductivity in mixture electrolyte systems. The authors curate and standardize data from two existing works, CALiSol and DiffMix, and augment the molecular representation by constructing geometric graphs where atoms are characterized by both invariant features and equivariant features. Then the authors develop GeoMix which maintains Set-SE(3) equivariance while capturing fine-grained geometric relationships between molecules, overcoming the limitations of existing methods. The extensive experimental results demonstrate the superior performance of GeoMix over baseline models, demonstrating its effectiveness in modeling mixture systems.

**Questions:**

1. GeoMix reports mean squared error (MSE = 0.035), while DiffMix reports mean absolute error (MAE = 0.044), which corresponds to an estimated MSE of around 0.002 by a rough estimation. While both the two paper present near perfect regression based on the true values vs predicted values plot, could this be due to differences in normalization, target scaling, or task formulation, and how should we interpret these metrics fairly across the two papers?

1. The GeoMix paper uses a different data split strategy from the original DiffMix paper, despite DiffMix providing the random seed for reproducibility. Therefore, is any reason for not using the same data split as DiffMix which could have been included as a baseline model?

1. GeoMix generates mixture geometries by randomly translating and rotating molecular conformers, forming mixture-level graphs from a single such placement. Is there any investigation in how sensitive the model is to this sampling strategy?

**Ethical Concerns:**

["NO or VERY MINOR ethics concerns only"]

**Final Justification:**

After discussion with the authors, their rebuttal has resovled most of my concerns, especially the good generalization performance to temperatures. Meanwhile, they also acknowledged the importance for more physics-consistent data sampling strategies, and they explained why a direct comparison to DiffMix is difficult. Therefore, I raised my score accordingly.

**Limitations:**

Yes.

**Quality:**

3

**Strengths And Weaknesses:**

Strengths:
1. The paper addresses an important and underexplored problem in machine learning for chemistry: predicting electrolyte properties from complex mixtures with geometric and compositional diversity. By introducing a mixture-level graph that includes atomic interactions across multiple molecules, the authors move beyond traditional molecular featurization approaches that treat molecules as independent graphs.

1. The paper introduces Geometric Interaction Network (GIN) which models the molecular interactions by intermolecular geometric message passing, surpassing previous methods which only learn a global feature vector of molecules while overlooking fine-grained and full-atom interactions across molecules.

1. The denoise loss adds a regularization that encourages robustness to small geometric perturbations, which is especially relevant in modeling systems with inherent thermal fluctuations and conformational noise.

Weaknesses:
1. Unlike DiffMix, which fits a VFT equation to enforce physically consistent trends across temperature, GeoMix treats temperature as a scalar input without enforcing any physical constraints. As a result, predictions at different temperatures may be inconsistent.

1. GeoMix constructs a mixture-level graph by placing individual molecules randomly in 3D space and connecting atoms based on proximity. However, the relative positions of molecules in a mixture are governed by thermodynamic distributions and local structuring (e.g., solvation shells), which are not captured by this random placement. Without sampling from realistic ensembles, such as those obtained via molecular dynamics, the constructed graphs may not reflect statistically meaningful intermolecular interactions.

---

> ### Author Rebuttal · Authors · 2025-07-31
>
> Thank you very much for your insightful and constructive feedback. We sincerely appreciate your careful review and the valuable suggestions you have provided. Below, we address your comments point by point.
>
> > **R1 (W1). Generalization at different temperatures.**
>
> We appreciate your insightful comment. Indeed, GeoMix does not enforce explicit physical constraints such as the VFT or Arrhenius equations during training. These constraints are typically adopted when the goal is to extrapolate conductivity at room temperature from high-temperature measurements, particularly in the context of molecular dynamics simulations. In contrast, our framework is designed as a general-purpose model applicable to various downstream tasks and properties, where hard-coding temperature-dependent functional forms may reduce flexibility or limit generalization.
>
> That said, we recognize the importance of ensuring physical consistency with respect to temperature. To empirically assess this, we performed an OOD generalization experiment where we trained the model on data with temperature ≤ 320 K (using a train/valid split of 4:1), and evaluated on samples with temperature > 320 K. Results show that GeoMix achieves significantly better predictive performance on this challenging temperature extrapolation task compared to other baseline models, suggesting that the learned representations effectively capture physically meaningful temperature trends without requiring hard-coded constraints.
>
> **Table S1: Results of OOD evaluation on high temperature.**
>
> |Models|MSE|MAE|Pearson r|Spearman r|
> |--|--|--|--|--|
> |MLP|32.43188|3.85718|0.43183|0.55550|
> |MM_MoLFormer|22.93524|2.83730|0.51219|0.56041|
> |MolSets_conv|9.92957|2.19592|0.83933|0.84519|
> |MolSets_SAGE|9.19274|2.13546|0.78423|0.80184|
> |EGNN_atten|8.13370|1.59404|0.81299|0.86055|
> |TFN_atten|6.51550|1.38267|0.85262|0.88674|
> |EGNN_linear|7.67502|1.61563|0.82692|0.84713|
> |TFN_linear|4.77976|1.28618|0.91367|0.92965|
> |GeoMix_EGNN|2.36597|**0.88293**|0.94950|**0.94805**|
> |GeoMix_TFN|**2.35390**|0.91722|**0.95186**|0.94535|
>
> > **R2 (W2). Geometric interactions without MD simulations.**
>
> Thank you for this important observation. We fully recognize that Molecular Dynamics (MD) simulations offer physically realistic ensembles that capture crucial thermodynamic distributions and local structuring, such as solvation shells. However, generating large, diverse MD-based datasets is highly challenging due to substantial computational costs and the scarcity of robust force fields for complex electrolyte mixtures. A key motivation of our work is to develop a learning-based alternative capable of capturing meaningful intermolecular interactions without relying on resource-intensive MD simulations.
>
> Although our graph construction does not impose physical constraints on initial molecular placement, the model architecture is specifically designed to recover relevant spatial relationships from the data. The Geometry Interaction Network (GIN) enables atomic-level message passing across molecules, allowing the model to learn interaction patterns reminiscent of those observed in MD ensembles. The learnable transformation matrices for each atom pair further serve as flexible approximations of local rearrangements, such as solvation and steric effects. Additionally, the use of Noisy Nodes introduces a form of structural noise, which encourages robustness and helps the model generalize to the kinds of fluctuations typically seen in MD simulations.
>
> We agree that integrating more physically realistic configurations would further enhance model fidelity, and as part of future work, we plan to explore incorporating MD-derived structures or Radial Distribution Functions (RDFs), potentially as auxiliary supervision. This potential extension will be discussed in the revised manuscript.
>
> > **R3 (Q1 & Q2). Fairness compared to DiffMix.**
>
> Thank you very much for your insightful comments regarding the comparison between GeoMix and DiffMix, especially in relation to evaluation metrics and dataset split strategies.
>
> First, we would like to clarify that a direct, numerical comparison of performance between GeoMix and DiffMix is not straightforward, primarily due to fundamental differences in task formulation and dataset construction. Specifically, the two works differ in the distributions of target values, normalization procedures, and data split schemes. Additionally, DiffMix incorporates supplementary features such as excess molar enthalpy and volume—features that are not used in GeoMix—which may contribute to their reported regression performance.
>
> Second, with respect to dataset splits, although the DiffMix repository provides an overall 8:1:1 split ratio (train:validation:test), it does not release the specific split indices or the individual 3D molecular structures used. As a result, it is challenging to reproduce their data partitions and baseline models without access to these intermediate processed datasets. In contrast, our study employs a consistent and standardized 7:2:1 split for both the CALiSol-23 and DiffMix datasets, facilitating fair within-study comparison and robust cross-dataset validation. We believe this approach ensures methodological consistency throughout our work.
>
> For these reasons, metric-to-metric comparisons are inherently limited due to disparities in data preprocessing, feature design, and splitting strategies. Moving forward, we support community-driven efforts to unify benchmark standards and facilitate the sharing of intermediate representations, which will enable more reliable and consistent comparisons across different models.
>
> Thank you again for highlighting these important considerations.
>
> > **R4 (Q3). Misunderstanding of random translation and rotation.**
>
> Thank you for highlighting this important aspect. We apologize for the misunderstanding and appreciate the opportunity to clarify this point. Our approach does NOT involve applying any random translations or rotations to molecular conformers. Instead, each molecule is first transformed into a canonical coordinate frame using PCA, after which the hybrid-level geometry is deterministically constructed. Additionally, our model is SE(3)-equivariant with respect to the input molecular conformers. As a result, the outputs are insensitive to random translations and rotations, as mentioned in your question.
>
> ***
>
> Thank you again for your thoughtful and detailed review. We greatly value your feedback and will address all points raised by incorporating clarifications and new analyses into the revised manuscript. We hope these efforts demonstrate the strength of our work and that you will reassess our contributions.

---

> > ### Comment · Reviewer_Syf6 · 2025-08-06
> >
> > Thank the authors for the response. The prompt and comprehensive response has addressed most of my concerns. Although I still think the physics-consistency in prediction and a direct comparison to DiffMix matters, the authors clearly explain the generalization performance to temperatures and the difficulty in reproducing the DiffMix results. Therefore, I'd like to raise my score.

---

> > > ### Author Response · Authors · 2025-08-06
> > >
> > > Dear Reviewer,
> > >
> > > We are very pleased that our response has helped clarify your questions. Your insightful comments and suggestions have greatly enhanced the quality and depth of our manuscript. It has been a pleasure to engage in this discussion with an expert of your multidisciplinary background, and we will incorporate all of our exchanges into the revised manuscript. Thank you again for your time, effort, and valuable insights.
> > >
> > > Best regards, The Authors

---

> ### Author Response · Authors · 2025-08-04
>
> Dear Reviewer,
>
> We sincerely appreciate your insightful and constructive feedback, which has significantly improved our work.
>
> We respectfully hope that, in light of these clarifications and improvements, you may consider reassessing the contribution and significance of our work. If you have any further questions or suggestions, we would be very grateful for the opportunity to engage in further discussion.
>
> Thank you once again for your time and thoughtful feedback.
>
> Best regards, The Authors

---

### Official Review · Reviewer_FnmS · 2025-07-01

**Clarity:** 3
**Significance:** 3
**Originality:** 4
**Rating:** 5
**Confidence:** 2

**Summary:**

This paper introduces GeoMix, a geometry-aware framework for predicting ionic conductivity in multi-molecule electrolyte mixtures. GeoMix first augments two public datasets (CALiSol and DiffMix) with geometric graphs, then couples an intra-molecular equivariant backbone (EGNN or TFN) with a novel Geometric Interaction Network (GIN) that passes messages between molecules through learned rotation/translation operators that respect Set-SE(3) symmetry. Across both datasets, GeoMix-TFN reduces test-set MSE by a large margin relative to the strongest geometric baselines, establishing new state-of-the-art performance.

**Questions:**

1. "For molecules not found in PubChem, we generate 3D structures 246 using RDKit [29] or download conformations from the Materials Project [30]." -- How do you handle cases where a single molecule has multiple low-energy conformers? Do you average over them, select a representative structure, or treat each conformer as a separate training sample?

2. How does GeoMix scale with mixture size M and total atoms N in practice (time/GB)? What is the asymptotic time/space complexity of the GIN with respect to M and N?

3. In Eq. (3) you state that the symmetry is defined such that each molecule is independently SE(3)-equivariant. Could you clarify why independent equivariance is reasonable? In realistic systems, inter-molecular interactions can couple the components.

**Ethical Concerns:**

["NO or VERY MINOR ethics concerns only"]

**Final Justification:**

The authors' response has satisfactorily addressed my concerns. I believe this work is significant for the NIPS conference and the entire field. I have raised my score accordingly.

**Limitations:**

Yes, the authors have discussed their limitations.

**Paper Formatting Concerns:**

No.

**Quality:**

3

**Strengths And Weaknesses:**

Strengths:
+ Strong theoretical underpinning and clear derivations.
+ Large performance gains on two datasets with code released.
+ Careful ablations isolate the contribution of each design choice.

Weaknesses
+ **Dataset Curation and Conformational Noise**. One potential concern is the dataset curation step, which may introduce conformational noise, especially when molecules with multiple low-energy conformers are considered. The paper does not discuss how multiple conformers are treated—whether they are averaged, a representative structure is selected, or each conformer is treated as a separate training sample. Does this affect the quality of the dataset?
+ **Complexity analysis absent.** The paper provides an in-depth explanation of the network architecture, but there is no discussion of the computational complexity of the GeoMix framework in terms of time and memory consumption. With potentially high-order backbone networks, what is the complexity of the overall proposed method?

---

> ### Author Rebuttal · Authors · 2025-07-31
>
> Thank you very much for your positive feedback and thoughtful, detailed comments. We greatly appreciate your careful reading and constructive suggestions, which will help us improve both the clarity and impact of our paper. We summarize and respond to your questions as follows:
>
> > **R1 (W1 & Q1). Dataset curation and conformational noise.**
>
> Thank you for raising this important point regarding conformational noise and dataset curation. In our current pipeline, we assign a single representative 3D structure to each molecule: if available, we retrieve a 3D conformation from PubChem or the Materials Project and optimize it using RDKit’s force field; otherwise, we generate and optimize the structure directly with RDKit. This approach ensures consistency across all data samples. Selecting a single conformer is a common convention in molecular machine learning, but our framework is compatible with alternative protocols, such as including multiple conformers as independent samples, should such data become available in the future.
>
> Here, to better evaluate model robustness to conformational variability, we have conducted an additional experiment: we apply Gaussian perturbations ($\sigma=0.3$ Å) to atomic positions, simulating conformational diversity. As shown in **Table S1** below, most models—including GeoMix_EGNN—exhibited only a marginal performance drop, indicating stability under moderate geometric deviations. Notably, the TFN encoder derives much worse performance, likely due to its reliance on higher-degree equivariant features (e.g., spherical harmonics) that could be sensitive to atomic position perturbations. We will make this limitation explicit in the revised manuscript.
>
> **Table S1: Results of coordinate perturbation test.**
> |Models|MSE|MAE|Pearson r|Spearman r|
> |--|--|--|--|--|
> |MolSets_conv|1.32605|0.70309|0.95992|0.96488|
> |MolSets_SAGE|1.74835|0.79402|0.94599|0.95805|
> |EGNN_atten|1.91948|0.58744|0.94193|0.96495|
> |TFN_atten|2.29420|0.75980|0.92930|0.95379|
> |EGNN_linear|1.11415|0.55469|0.96660|0.97623|
> |TFN_linear|2.39326|0.93700|0.92892|0.94426|
> |GeoMix_EGNN|**0.63117**|**0.38836**|**0.98174**|**0.98348**|
> |GeoMix_TFN|4.51728|1.32373|0.85851|0.88631|
>
> We appreciate your suggestion and will include these clarifications and perturbation results in the final version.
>
> > **R2 (W2 & Q2). Complexity analysis absent.**
>
> Thank you for raising the important issue of computational complexity and scalability.
>
> The main computational cost in GeoMix comes from two sources: (1) the molecular encoder, which processes individual molecules, and (2) the cross-molecular interaction module (GIN), which performs global message passing between all atoms across different molecules. For a mixture containing a total of $N$ atoms (typically fewer than 100 per sample), these components scale as follows:
> - **Molecular Encoder:** For EGNN, the per-layer time complexity is $\mathcal{O}(N^2)$. For TFN, the per-layer complexity rises to $\mathcal{O}(N^2\cdot L^6)$ with $L$ being the maximum tensor order used, due to the involvement of higher-degree features.
> - **GIN Module:** The cross-molecular interaction is modeled as a fully connected bipartite graph (across molecular pairs), also resulting in an $\mathcal{O}(N^2)$ time and space complexity per layer, typically dominating total costs.
>
> In practice, as $N$ is usually modest, the computational cost remains tractable even for deeper networks or high-degree backbones. Although our method scales quadratically with the number of atoms, it remains significantly more efficient than traditional molecular dynamics simulations.
>
> We have also benchmarked inference time and memory usage on the CALiSol dataset. Results are shown in **Table S2** below.
> - The inference time of GeoMix is comparable to that of baseline models with attention-based encoders (e.g., EGNN_atten, TFN_atten). In these models, the attention mechanism, rather than our interaction module, is typically the main contributor to the runtime cost.
> - GeoMix’s memory usage is higher due to modeling all pairwise atom-level edges across molecules, but this overhead correlates with a clear improvement in predictive performance.
>
> **Table S2: Runtime analysis on CALiSol dataset.**
>
> |Models|Inference Time ($\times 10^{-2}$s)|Memory (GB)|MSE|Pearson r|
> |--|--|--|--|--|
> |EGNN_linear|4.41|2.20|1.461|0.951|
> |TFN_linear|24.13|5.91|1.107|0.967|
> |EGNN_atten|23.40|2.22|2.666|0.908|
> |TFN_atten|43.20|5.91|1.808|0.946|
> |GeoMix_EGNN|34.04|25.15|0.552|0.985|
> |GeoMix_TFN|53.64|45.18|**0.432**|**0.987**|
>
> > **R3 (Q3). Rationality of independent SE(3)-equivariant.**
>
> Thank you for this insightful question regarding independent SE(3)-equivariance.
>
> Our statement refers specifically to the initial geometric encoding stage, where each molecule is provided as an independent input with its own atomic coordinates and frame of reference. This setup makes it mathematically valid to apply arbitrary rigid transformations to each molecule separately, justifying independent equivariant embeddings at this stage.
>
> However, our model explicitly accounts for intermolecular interactions in subsequent layers via the GIN module, which passes features and enables attention-based aggregation across molecules. These steps allow the model to capture coupling effects and realistic intermolecular behaviors, while still respecting fundamental symmetries.
>
> We also acknowledge the limitation that our model assumes molecules are not co-embedded in a shared spatial configuration—a reflection of experimental constraints, since such atomistic details are rarely available for mixtures in solution. Nonetheless, our design balances physical principles with practical considerations, ensuring that meaningful coupling can be learned even with independent geometric encodings. We will clarify this rationale in the revised manuscript.
>
> ***
>
> Thank you once again for your constructive feedback, which has directly contributed to improving the clarity, empirical evaluation, and theoretical discussion of our work. We will incorporate all clarifications and additional results into the revised version of the manuscript. We sincerely hope that our responses and revisions will address your concerns, and we would be grateful if you might consider these improvements in your evaluation.

---

> > ### Comment · Reviewer_FnmS · 2025-08-04
> >
> > Thanks for the response. The authors' response has satisfactorily addressed my concerns. I believe this work is significant for the NIPS conference and the entire field. I have raised my score accordingly.

---

> > > ### Author Response · Authors · 2025-08-04
> > >
> > > Dear Reviewer,
> > >
> > > Thank you very much for your recognition and support. Your encouragement is truly inspiring for our team. It has been our pleasure to address your questions, and we have greatly benefited from your thoughtful suggestions. We will incorporate these valuable discussions and improvements into our revised manuscript.
> > >
> > > Best regards, The Authors

---

### Official Review · Reviewer_1upL · 2025-07-01

**Clarity:** 3
**Significance:** 3
**Originality:** 3
**Rating:** 4
**Confidence:** 4

**Summary:**

This paper introduces GeoMix, a novel geometric framework for predicting properties of molecular mixtures, specifically focusing on electrolyte conductivity. The authors address two key challenges: (1) the lack of standardized benchmarks in the field, and (2) inadequate modeling of geometric interactions between molecules in mixture systems. They curate two datasets (CALiSol and DiffMix) and propose a Geometric Interaction Network (GIN) that maintains Set-SE(3) equivariance while enabling fine-grained intermolecular geometric message passing through learnable coordinate transformations.

**Questions:**

1. Why use PCA for frame construction? Have you tried other methods (e.g., based on molecular moments of inertia)? How does the model behave for e.g. spherical molecules (PC) where PCA axes are arbitrary?
2. Could you visualize some learned transformation matrices to see if they capture meaningful patterns? Does those transformations correlate with MD‑derived relative orientations?
3. Can the model extrapolate to mixtures with more components than training data?
4. Can you report “train on DiffMix, test on CALiSol” and vice‑versa, or a leave‑one‑solvent‑family‑out split?

**Ethical Concerns:**

["NO or VERY MINOR ethics concerns only"]

**Final Justification:**

Most of my concerns have been addressed, and the paper is well-written. However, given the limited technical novelty—the architecture is not new, and neither is the dataset—I would assign this paper a borderline accept score.
To strengthen the paper, I encourage the authors to include a visualization of the learned transformations, which represents one of the few technical novelties in this work. Additionally, the dataset limitations remain unaddressed. I suggest the authors include more diverse systems to substantiate their claim that "the GeoMix framework is designed to be chemically general and data-driven"

**Limitations:**

Yes.

**Quality:**

4

**Strengths And Weaknesses:**

**Strength**

1. **Benchmark Standardization**: The effort to curate and standardize the CALiSol and DiffMix datasets is commendable and provides significant value to the community. The careful preprocessing with geometric graph construction addresses a real need in the field.

2. **Novel Geometric Framework**: The learnable transformation approach between molecular coordinate systems is innovative. The mathematical formulation of Set-SE(3) equivariance for mixture systems is rigorous and well-motivated.

3. **Strong Empirical Results**: GeoMix consistently outperforms diverse baselines, achieving significant improvements (e.g., MSE of 0.432 vs 1.107 on CALiSol).

4. **Extensive Comparison on Learnable Transformation** Comparison with different learnable transformation (quaternions etc.) is interesting and commendable.

5. **Thorough ablations & theory appendix**: Clarifies why non‑orthogonal matrices increase expressivity and proves equivariance.

**Weakness**
1. **Limited Physical Interpretability**:  The physical meaning of  the learnable transformations is unclear. The paper doesn't discuss what these transformations represent  in terms of actual molecular interactions.

2. **Computational Complexity**: The method requires computing transformations for all atom pairs across molecules, which could be computationally expensive for large systems. The paper doesn't discuss scalability or computational costs. Furthermore, no runtime analysis or comparison with baselines.

3. **Dataset Limitations**: Generalisation outside Li‑ion solvents untested. Both datasets are Li‑salt carbonates/ethers; performance on, e.g., Na‑ion or ionic‑liquid systems is unknown.

---

> ### Author Rebuttal · Authors · 2025-07-31
>
> Thank you very much for your valuable and constructive feedback. We greatly appreciate your careful reading, positive remarks, and thoughtful questions. Please find our detailed responses below:
>
> > **R1 (W1 & Q2). Limited physical interpretability & visualization.**
>
> Thank you for your thoughtful question about the physical meaning of our model's learnable transformations. We appreciate the opportunity to better clarify their connection to molecular interactions.
>
> Our goal of introducing these transformation matrices and translation vectors is to model the relative geometry between interacting molecules in a flexible, data-driven way. This design reflects the fact that real electrolyte interactions depend on spatial arrangement, orientation, and solvation effects, which are often more complex than rigid SO(3) rotations. Allowing more general transformations lets the model learn subtle geometric patterns—such as solvation shell adjustment and local flexibility—that rigid constraints might miss. This is supported by our ablation studies (Section 4.4), which show that strictly enforcing SO(3) rotation reduces prediction performance.
>
> We agree that direct visualization of the learned transformations would enhance physical interpretability, especially in revealing correlations with molecular dynamics (MD)-derived orientations. While the current NeurIPS format limits the inclusion of visualizations during the discussion, we plan to incorporate such analysis in the revised version.
>
> > **R2 (W2). Computational complexity.**
>
> Thank you for raising the important point regarding computational complexity and scalability. You are correct that our method computes interactions between all intermolecular atom pairs, resulting in a complexity of $\mathcal{O}(N^2)$, where $N$ is the total number of atoms from one copy of each component molecule. In our datasets, $N$ is generally below 100, and this approach enables the model to capture detailed pairwise interactions crucial for accurate property prediction in molecular mixtures.
> As you suggested, we have added a runtime and memory analysis (see **Table S1**) comparing our method with several baselines on the CALiSol dataset. The results show:
> - The inference time of GeoMix is comparable to that of baseline models with attention-based encoders (e.g., EGNN_atten, TFN_atten). In these models, the attention mechanism, rather than our interaction module, is typically the main contributor to the runtime cost.
> - GeoMix’s memory usage is higher due to modeling all pairwise atom-level edges across molecules, but this overhead correlates with a clear improvement in predictive performance.
>
> We hope this additional analysis clarifies the scalability of our method. We believe the observed performance improvements justify the higher computational costs for most practical scenarios involving moderately sized molecular systems.
>
> **Table S1: Runtime analysis on CALiSol dataset.**
> |Models|Inference Time ($\times 10^{-2}$s)|Memory (GB)|MSE|Pearson r|
> |--|--|--|--|--|
> |EGNN_linear|4.41|2.20|1.461|0.951|
> |TFN_linear|24.13|5.91|1.107|0.967|
> |EGNN_atten|23.40|2.22|2.666|0.908|
> |TFN_atten|43.20|5.91|1.808|0.946|
> |GeoMix_EGNN|34.04|25.15|0.552|0.985|
> |GeoMix_TFN|53.64|45.18|**0.432**|**0.987**|
>
> > **R3 (W3). Dataset limitations.**
>
> Thank you for raising this important point regarding dataset diversity and generalization. We agree that evaluating model performance on other electrolyte systems, such as Na-ion or ionic liquids, would be valuable for understanding broader applicability.
>
> While our current study is limited to Li-salt electrolytes (carbonates and ethers), we would like to emphasize that the GeoMix framework is designed to be chemically general and data-driven. It operates on atom-level features and learns intermolecular geometric interactions via equivariant message passing, which should, in principle, capture fundamental physical interactions relevant across diverse electrolyte chemistries.
>
> GeoMix can be readily adapted to systems containing alternative cations (e.g., Na⁺, K⁺, or organic ions) or novel solvents, provided that datasets include consistent conductivity measurements and 3D molecular structures. Even for ionic liquids with more complex ion packing, the model's representational capability remains valid.
>
> However, we note that large-scale, well-annotated datasets for non-Li electrolytes with the necessary structural information are currently unavailable to our knowledge. We have explicitly acknowledged this limitation in the revised manuscript and hope our work will stimulate future data collection and benchmarking efforts for broader electrolyte chemistries.
>
> > **R4 (Q1). PCA frame constructions.**
>
> Thank you for your thoughtful question regarding our use of PCA for molecular frame construction. In response to your suggestion, we have evaluated an alternative based on the Molecular Moment of Inertia (MMI) tensor, where principal axes are defined as eigenvectors of the inertia matrix. We have now included this variant in our ablation study. The results are as follows:
>
> **Table S2: Results of ablation study about frame constructions.**
>
> |Models|MSE|Pearson r|
> |--|--|--|
> |GeoMix_EGNN (PCA Frame)|0.552|0.985|
> |GeoMix_EGNN (MMI Frame)|0.542|0.984|
>
> As shown, both methods yield nearly identical performance, suggesting that the model is largely insensitive to the specific frame construction method. This is expected since PCA and inertia-based axes are both computed through spectral decomposition (e.g., SVD) of second-order symmetric tensors (covariance matrix or inertia tensor), and therefore often produce similar or equivalent orthonormal bases for molecular structures.
>
> It is also important that our model treats molecular frames as fixed inputs determined prior to training. These frames are not learnable parameters and do not depend on model weights or optimization. The model's predictions are robust to the choice of frame construction method, as long as the method is deterministic and consistently applied, due to the equivariant nature of the geometric message-passing operators.
>
> Furthermore, molecules with high symmetry (e.g., spherical molecules) do not affect the behavior of our model. Although the selection of the local coordinate system may be arbitrary, all geometric interactions are computed within a standardized coordinate system, where the transformations $\boldsymbol{R}\_{ij}^{m,n},\boldsymbol{t}\_{ij}^{m,n}$ are invariant to the selected local frame. In practice, perfectly symmetric molecules are extremely rare except for isolated atoms.
>
> We hope this clarifies the rationale and robustness of our frame construction strategy. We will include a brief discussion and the additional experimental result in the revised version.
>
> > **R5 (Q3 & Q4). OOD evaluation of different dataset splitations.**
>
> Thank you for these valuable questions on the model's ability to extrapolate to mixtures with more components or unseen solvent families, and on cross-dataset generalization.
>
> While we have not yet performed the exact "train on DiffMix, test on CALiSol" experiments, we have designed a related experiment to directly probe the model's extrapolation to new mixture regimes. Specifically, we trained the model only on samples with conductivity ≤10 mS/cm (80/20 train/validation split), while all high-conductivity samples (>10 mS/cm) were reserved for testing. This protocol compels the model to predict for mixtures and composition spaces not seen during training, including those likely to contain more components, novel combinations, or distinct solvent–electrolyte behaviors.
>
> We believe this conductivity-based split is a stringent test of the model’s extrapolation to novel mixture compositions and more diverse regimes, beyond mere memorization of training patterns.
>
> **Table S3: Results of OOD evaluation on high-conductivity test set.**
>
> |Models|MSE|MAE|Pearson r|Spearman r|
> |--|--|--|--|--|
> |EGNN_linear|32.72010|4.45704|0.25309|0.34116|
> |TFN_linear|24.21760|3.58482|0.39704|0.50828|
> |GeoMix_EGNN|**17.13204**|**2.85305**|**0.57858**|0.57091|
> |GeoMix_TFN|19.42731|2.92452|0.43568|**0.60949**|
>
> The results (see **Table S3**) demonstrate that GeoMix, especially with the EGNN encoder, consistently outperforms baseline methods—even under this challenging setting.
>
> ***
>
> Thank you again for your comprehensive review and high-level comments, which have significantly helped us to improve the clarity, rigor, and impact of our work. We will incorporate all clarifications, additional experiments, and references as discussed in the revised manuscript.

---

> ### Author Response · Authors · 2025-08-04
>
> Dear Reviewer,
>
> Thank you for your detailed and thoughtful evaluation. Currently, we have addressed your questions in the previous response. Your feedback has significantly enhanced the clarity and rigor of our manuscript.
>
> We would be truly grateful for any further suggestions you may have on these clarifications or the additional results. Your insights have been invaluable in improving our work.
>
> Best regards, The Authors

---

> > ### Comment · Reviewer_1upL · 2025-08-04
> >
> > I thank the authors for their engagement. Most of my concerns have been addressed. I will, however, leave my score unchanged.

---

> > > ### Author Response · Authors · 2025-08-05
> > >
> > > Dear Reviewer,
> > >
> > > Thank you for your continued engagement and for taking the time to review our responses. We are grateful that most of your concerns have been addressed. Your feedback will be valuable in improving the clarity and rigor of our work. We sincerely appreciate your thoughtful input throughout the review process.
> > >
> > > Best regards, The Authors

---

### Official Review · Reviewer_x1YU · 2025-07-09

**Clarity:** 3
**Significance:** 2
**Originality:** 2
**Rating:** 4
**Confidence:** 4

**Summary:**

This paper proposes GeoMix, an equivariant neural network model for predicting the electrical conductivity of electrolyte solutions. A systematic comparison of GeoMix with various neural network models, including MLP-based, topology-based, and geometry-based architectures, is conducted on the task of electrical conductivity prediction. Furthermore, the study supplements the CALiSol-23 and DiffMix datasets with relevant information, enhancing the future evaluation of predictive capabilities on these datasets.

**Questions:**

See weaknesses.

**Ethical Concerns:**

["NO or VERY MINOR ethics concerns only"]

**Final Justification:**

My primary concern stemmed from the experiments in the paper, which the authors have since addressed by supplementing additional experiments in the rebuttal. The writing and experimental evaluation of this paper meet my threshold for acceptance. However, I must emphasize that the reported Pearson R of 0.58 is not practically useful for scientists, which limits the practical applicability of this work.

**Limitations:**

yes

**Quality:**

3

**Strengths And Weaknesses:**

## Strengths

This work introduces a novel equivariant 3D graph neural network model for predicting the electrical conductivity of mixed molecular systems. The proposed model achieves state-of-the-art (SOTA) prediction accuracy on both the CALiSol-23 and DiffMix datasets.

## Weaknesses

**Major issues**

1. The evaluation of electrical conductivity prediction accuracy in this work relies solely on random splitting within the CALiSol-23 and DiffMix datasets. This approach risks yielding overly optimistic results, as evidenced by the Pearson r exceeding 0.98 for GeoMix and generally above 0.95 for baseline models in Table 1. In practical electrical conductivity prediction (and molecular property prediction in general), models frequently encounter out-of-sample scenarios, such as predicting the conductivity of: unknown electrolytes in known solvents, known electrolytes in unknown solvents, or novel combinations of both unknown electrolytes and solvents. Therefore, it is recommended that the authors explicitly designate specific electrolyte and solvent molecules to be held out exclusively in test set for evaluating the model's predictive performance under these out-of-sample conditions.

2. Predicting electrolytes with higher conductivity is a critical scientific objective. To assess this capability, it is recommended that the authors partition a subset of high-conductivity data as an independent test set, utilizing only low-conductivity data for training and validation. Specifically, for the CALiSol-23 dataset, data points with conductivity >20 mS/cm could be reserved solely for testing. Similarly, for the DiffMix dataset, data points with conductivity >10 mS/cm could be reserved for testing. This strategy would provide a meaningful assessment of the model's extrapolation performance.

3. Upon reviewing the provided data and code, it was observed that for all lithium salts, only the 3D conformations of their anionic moieties were generated. While this approach might be adequate for the current dataset containing solely lithium cations, its applicability becomes significantly limited when datasets incorporate other cations, thereby severely constraining the model's generalization ability. Although generating accurate 3D structures for complete ionic compounds (containing both cation and anion) presents challenges, the authors should nevertheless consider and address how to effectively integrate the structural features of diverse cations into the model.

**Minor issues**

1. The term "DiffMix" is used to refer to both a dataset and a model in the manuscript, which may cause confusion. It is recommended to distinguish them by adding prefixes or suffixes

2. The authors' expansion of the CALiSol-23 and DiffMix datasets appears limited to supplementing 3D molecular coordinates (without DFT optimization and excluding cation structures for ionic compounds). If accurate, the claims regarding dataset contributions should be moderated.

3. Although no implementation is available for the DiffMix model, its original publication reports a conductivity prediction MAE of 0.044 mS/cm, a result comparable to or potentially better than the best performance in this work. The authors should compute and report GeoMix's MAE on the DiffMix dataset to enable direct comparison.

4. The page number of reference 10 is incorrect, and the issue number of reference 11 is redundant ('(1)' should be deleted). Please check the accuracy of all references thoroughly.

5. In the Checklist section, the author seems to have provided two code repositories, https://anonymous.4open.science/r/GeoMix-74AC/ and https://anonymous.4open.science/r/GeoMix-73AC/. The former is an invalid address. Please correct it.

---

> ### Author Rebuttal · Authors · 2025-07-31
>
> Thank you for your thoughtful comments and questions.  We truly appreciate the time and effort you invested in reviewing our work.  Below, we summarize and respond to your points in detail.
>
> > **R1 (Major Issue 1). The necessity of out-of-sample evaluation.**
>
> Thank you very much for your insightful comments regarding the evaluation protocol and the importance of testing the model’s performance under out-of-sample scenarios. We fully agree that for practical electrical conductivity prediction tasks, it is critical to assess how well a model generalizes to completely new combinations of electrolytes and solvents, as well as to previously unseen individual molecules.
>
> Indeed, we recognize that extrapolating to completely novel electrolytes or solvents remains a significant challenge, particularly given the limited size and chemical diversity of current datasets. In realistic deployment scenarios, it is often more plausible to assume that the model has previously encountered the individual constituent molecules (both solvents and electrolytes) during training, though possibly only in combinations of different concentration ratios. Our current experimental design adopts this assumption and primarily focuses on interpolation within relevant composition spaces, which is crucial for accurately modeling mixture behaviors.
>
> Nonetheless, we fully acknowledge the importance of evaluating model robustness beyond random data splits. In response to the reviewer’s valuable suggestion, we have now conducted an additional evaluation in which the dataset is partitioned based on conductivity ranges, rather than through random sampling. This protocol provides a meaningful proxy for assessing out-of-sample generalization, particularly with respect to extrapolation from low- to high-conductivity regimes. The detailed results and analysis of this experiment are presented in our response **R2** below.
>
> > **R2 (Major Issue 2). Out-of-sample evaluation of high-conductivity electrolytes.**
>
> Thank you for your recommendation to hold out high-conductivity electrolytes as a more meaningful out-of-sample test set. In line with your suggestion, we have performed new experiments on the CALiSol-23 dataset, where we use only samples with conductivity ≤10 mS/cm for training and validation (random split of 8:2), and reserved all samples with conductivity >10 mS/cm exclusively for testing.
>
> We would like to note that while we initially consider using the >20 mS/cm threshold as you proposed, there are only 64 samples in this range, which is insufficient for statistically robust evaluation. To ensure a meaningful and reliable assessment, we therefore adopt the >10 mS/cm threshold, resulting in a high-conductivity test set comprising 1,244 data points. This experimental protocol thus enables a direct and statistically sound evaluation of the model’s capability to extrapolate to high-conductivity out-of-sample regimes.
>
> **Table S1: Results of out-of-sample evaluation on high-conductivity test set.**
>
> |Models|MSE|MAE|Pearson r|Spearman r|
> |--|--|--|--|--|
> |EGNN_linear|32.72010|4.45704|0.25309|0.34116|
> |TFN_linear|24.21760|3.58482|0.39704|0.50828|
> |GeoMix_EGNN|**17.13204**|**2.85305**|**0.57858**|0.57091|
> |GeoMix_TFN|19.42731|2.92452|0.43568|**0.60949**|
>
> As shown in the table, performance across all models decreases notably in this challenging out-of-sample setting, which underscores the difficulty of extrapolating to high-conductivity regimes. Importantly, our proposed GeoMix model, particularly with the EGNN encoder, demonstrates clear improvements over baseline approaches in all metrics. These results and discussions will be incorporated into our revised manuscript.
>
> > **R3 (Major Issue 3). Discussion when datasets incorporate other cations.**
>
> We appreciate your close reading of our code and data, and your attention to the implications for model generalization when datasets include cations other than lithium. We apologize for not explaining the relevant chemical background more clearly in the manuscript.
> In this work, we generate 3D conformations only for the anionic moieties, treating Li⁺ as a point charge without explicit geometry. This approach reflects the chemical reality of our dataset: the salts are fully dissociated, with cations and anions freely distributed, making it inappropriate to assign a fixed 3D structure to the complete ionic compound.
> Moreover, our model is general by design and can accommodate diverse cation types. While the current dataset only includes lithium salts (Li⁺), the proposed framework can readily integrate other cations (such as Na⁺, K⁺, or multivalent ions) by explicitly incorporating their structural and physicochemical features into the model input.  We will clarify this point in the revised manuscript.
>
> > **R4 (Minor Issue 2). Clarification of our dataset contributions.**
>
> We appreciate the reviewer’s careful reading.  Our contributions to the CALiSol and DiffMix datasets primarily consist of the following three aspects: (1) supplementing 3D molecular coordinates using standardized procedures such as RDKit conformer generation and PubChem retrieval to enable spatial graph learning;  (2) performing data cleaning by removing invalid or inconsistent samples to improve dataset quality and robustness;  and (3) establishing a standardized data split protocol to facilitate fair and reproducible benchmarking.
>
> > **R5 (Minor Issue 3). Clarifying DiffMix baseline comparison.**
>
> Thank you very much for your insightful comments regarding the comparison between GeoMix and DiffMix, especially in relation to evaluation metrics and dataset split strategies.
>
> First, we would like to clarify that a direct, numerical comparison of performance between GeoMix and DiffMix is not straightforward, primarily due to fundamental differences in task formulation and dataset construction. Specifically, the two works differ in the distributions of target values, normalization procedures, and data split schemes. Additionally, DiffMix incorporates supplementary features such as excess molar enthalpy and volume—features that are not used in GeoMix—which may contribute to their reported regression performance.
>
> Second, with respect to dataset splits, although the DiffMix repository provides an overall 8:1:1 split ratio (train:validation:test), it does not release the specific split indices or the individual 3D molecular structures used. As a result, it is challenging to reproduce their data partitions and baseline models without access to these intermediate processed datasets. In contrast, our study employs a consistent and standardized 7:2:1 split for both the CALiSol-23 and DiffMix datasets, facilitating fair within-study comparison and robust cross-dataset validation. We believe this approach ensures methodological consistency throughout our work.
>
> For these reasons, metric-to-metric comparisons are inherently limited due to disparities in data preprocessing, feature design, and splitting strategies. Moving forward, we support community-driven efforts to unify benchmark standards and facilitate the sharing of intermediate representations, which will enable more reliable and consistent comparisons across different models.
>
> Thank you again for highlighting these important considerations.
>
> > **R6 (Minor Issues 1, 4 & 5).**
>
> We sincerely thank the reviewer for the detailed and constructive feedback. Below we respond to each point individually:
> - **Minor Issue 1. Clarification of the term "DiffMix":** We agree that using "DiffMix" to refer to both the model and the dataset may cause confusion. In the revised version, we will clearly distinguish between the two by using "DiffMix-dataset" for the data and "DiffMix" for the baseline method throughout the manuscript and figures.
> - **Minor Issue 4. Reference formatting issues:** We have carefully checked all reference entries and will correct these formatting issues in the revised version.
> - **Minor Issue 5. Invalid code repository link:** We will remove the incorrect link and ensure only the valid repository is listed in the Checklist and manuscript.
>
> ***
> We sincerely appreciate your valuable suggestions and constructive feedback. We will incorporate the requested clarifications and additional results in the revised manuscript. We hope that our responses and the improvements made will allow you to reconsider the significance of our contributions. Thank you again for your time and thoughtful review.

---

> > ### Comment · Reviewer_x1YU · 2025-08-05
> >
> > We appreciate the additional experimental results provided during revision, which have addressed most of our concerns. Accordingly, we have raised our score. However, it must be emphasized that the current model still demonstrates unsatisfactory performance in out-of-sample evaluation (Pearson R 0.58, with an implied R² likely below 0.4), severely limiting its practical applicability. That said, the study tackles a highly meaningful problem, though substantial gaps remain before this problem can be truly resolved.

---

> > > ### Author Response · Authors · 2025-08-05
> > >
> > > Dear Reviewer,
> > >
> > > We sincerely thank you for your time and thoughtful feedback. We are pleased that the additional experimental results have addressed most of your concerns. Out-of-sample prediction and practical applicability remain critical challenges in this domain, and we are committed to further exploring strategies to enhance the generalization ability of the model in future work. Your constructive suggestions will be incorporated into the revised manuscript.
> > >
> > > Best regards, The Authors

---

> ### Author Response · Authors · 2025-08-04
>
> Dear Reviewer,
>
> Thank you for your detailed and thoughtful review. We have now posted a full response to your comments. In particular, we would be grateful if you could kindly reconsider your evaluation of our manuscript based on our analysis and explanations.
>
> If there are any remaining questions or concerns, we would be glad to provide further clarification. We sincerely hope that our revisions have addressed your concerns and that the improvements we made will be favorably considered during your re-evaluation.
>
> Thank you again for your time and consideration.
>
> Best regards, The Authors

---

### Author Response · Authors · 2025-08-09
**Rebuttal Acknowledgment**

Dear Reviewers and Area Chair,

We sincerely thank you for your time, effort, and insightful feedback on our manuscript.  Your constructive comments have greatly contributed to improving the clarity and quality of our work.

Through our discussions and the reviewer's responses, we are encouraged to see that the major concerns raised by all reviewers have been addressed.  Following the rebuttal, 3 out of 4 reviewers (Reviewer x1YU, Reviewer FnmS, and Reviewer Syf6) raised their ratings, and now the work has received uniformly positive judgment.  Notably, Reviewer FnmS found the work "is significant for the NIPS conference and the entire field," and Reviewer x1YU commented that the study "tackles a highly meaningful problem."  We are deeply grateful for the reviewers’ thoughtful suggestions and strong support.  All comments will be carefully considered and incorporated into the revised manuscript.

In this work, we first define the symmetries satisfied by mixture systems through Set-SE(3) equivariance.  We introduce GeoMix, a novel framework for mixture property prediction that preserves Set-SE(3) equivariance.  Its core module, GIN, enables full-atom equivariant message passing across molecules in mixture systems, allowing the model to implicitly learn geometric relationships between molecules.  To facilitate further research in this domain, we also curated and released two benchmarks of liquid electrolyte conductivity prediction.  All code and datasets have been made publicly available to promote transparency and reproducibility.

Once again, we thank you for your valuable suggestions and careful reviews.  Your feedback is instrumental in refining our contributions.  Thank you!

Best regards, The Authors

---

### Decision · Program_Chairs · 2025-09-17

**Decision:**

Accept (poster)

**Comment:**

Reviews of this work are positive, with reviewers praising the combination of data-centric (improved curation of CALiSol and DiffMix with geometric properties) and methodological contributions (various ways of incorporating both non-spatial-type molecular features such as atomic numbers, and spatial-type features) which build on the data made available by the authors. Different reviewers focused on different parts of these contributions (Reviewer x1YU and Reviewer Syf6 on the methodological side, Reviewer 1upL and Reviewer FnmS on the data side), but unanimously found the problem to be important and evaluation to be reasonably comprehensive. Most weaknesses raised had to do with computational costs of the methodological work not being discussed in enough detail, which the authors note are similar to other equivariant networks.

On basis of these reviews, I find the work has merit, and recommend acceptance.